

# Jet diffusion versus JetGPT – Modern networks for the LHC

Anja Butter[1,2], Nathan Huetsch[1], Sofia Palacios Schweitzer[1],
Tilman Plehn[1], Peter Sorrenson[3] and Jonas Spinner[1]

**1** Institute for Theoretical Physics, Universität Heidelberg, Germany
**2** LPNHE, Sorbonne Université, Université Paris Cité, CNRS/IN2P3, Paris, France
**3** Heidelberg Collaboratory for Image Processing, Universität Heidelberg, Germany

## Abstract

We introduce two diffusion models and an autoregressive transformer for LHC physics simulations. Bayesian versions allow us to control the networks and capture training uncertainties. After illustrating their different density estimation methods for simple toy models, we discuss their advantages for Z plus jets event generation. While diffusion networks excel through their precision, the transformer scales best with the phase space dimensionality. Given the different training and evaluation speed, we expect LHC physics to benefit from dedicated use cases for normalizing flows, diffusion models, and autoregressive transformers.

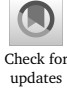

# 1 Introduction

The future of LHC physics lies in a systematic and comprehensive understanding of all aspects of the recorded data in terms of fundamental theory. This can be achieved through simulation-based analyses, applying and adapting modern data science methods. As obvious from the name, this method relies on a fast and precise simulation chain, starting with the hard scattering evaluated for a given Lagrangian, to fast detector simulations. Because LHC physics is driven by fundamental questions, these simulations have to be based on first principles, mere modeling would not allow us to extract relevant or interesting information from the data. Moreover, for theory predictions to not become a limiting factor to the entire LHC program, this simulation chain has to be (i) precise, (ii) fast, and (iii) flexible.

Modern machine learning (ML) has shown great potential to transform LHC analyses and simulations [1, 2]. The leading ML-tools for LHC simulations are, obviously, generative networks, which combine unsupervised density estimation over phase space with a fast sampling into the learned density. The list of tasks where (generative) neural networks can improve LHC simulations is long [1]. It starts with phase space integration [3, 4] and phase space sampling [5–9] of ML-encoded fast transition amplitudes [10, 11]. More advanced tasks include event subtraction [12], event unweighting [13, 14], or super-resolution enhancement [15, 16]. Prototypical applications which allow for a systematic evaluation of the network performance are NN-event generators [17–22], NN-parton showers [23–28], or detector simulations [29–45]. Even when trained on first-principle simulations, such fast generators are easy to handle, efficient to ship, and powerful in amplifying statistical limitations of the training samples [46, 47].

Classical generative network architectures include variational autoencoders (VAEs) and generative adversarial networks (GANs). Both of them can generate high-dimensional data, assuming that the intrinsic dimensionality of the problem is much smaller than the apparent dimensionality of its representation. However, both have not been shown to fulfill the precision requirements of the LHC. Precise density estimation points to bijective generative networks, for instance normalizing flows [48–52] and their invertible network (INN) variant [53–55], which are limited to lower-dimensional sampling but sufficient at least for the hard process at the LHC.

LHC studies are consistently showing promising results for normalizing flows,[1] including transformative tasks, like probabilistic unfolding [56–60], inference from high-dimensional data [61], or the matrix element method [62]. One reason why INNs have established a new level of stability, control and uncertainty estimation, is the combination with Bayesian neural network (BNN) concepts [63–69], discriminator-training and reweighting [70, 71], and conditional training on augmented data. In the spirit of explainable AI, Bayesian generative networks allow us to understand how networks learn a given phase space distribution, in the case of INNs very similar to a traditional fit [69]. They systematically improve the precision of the underlying density estimation and track the effects from statistical and systematic limitations of the training data [22]. In this study we will first compare the successful INNs with new diffusion networks [72–76] on the task of unconditional phase space generation. This allows us to benchmark their performance as surrogate simulators [22, 46, 47], as well as prepare their future usage for conditional tasks such as probabilistic unfolding [56–60].

An aspect of neural networks which is often overlooked is that in precision LHC simulations the intrinsic dimension of the physics problem and the apparent dimension of its phase space are similar; for this dimensionality we need to encode all correlations [77, 78]. This implies that the network size, its training effort, and its performance tend to scale poorly with the number of particles and suffer from the curse of dimensionality. This is the motivation to

---

[1]Note that in these applications autoregressive flows do *not* outperform advanced coupling layers.

also include an autoregressive [79, 80] transformer [81] in our study of modern generative networks.

In this paper we will introduce two new different diffusion models for particle physics applications in Secs. 2.1 and 2.2. We then introduce a new autoregressive, eventually pre-trained, transformer architecture (JetGPT) with an improved dimensionality scaling in Sec. 2.3. For all three networks we develop new Bayesian versions, to control their learning patterns and the uncertainty in the density estimation step. In Sec. 3 we illustrate all three models for two toy models, a two-dimensional linear ramp and a Gaussian ring. Finally, in Sec. 4 we use all three networks to generate $Z$+jets events for the LHC, the same reference process as used for uncertainty-aware INNs in Ref. [22]. This standard application allows us to quantify the advantages and disadvantages of the three new architectures and compare them to the INN benchmark.

## 2 Novel generative networks

At the LHC, generative networks are used for many simulation and analysis tasks, typically to describe virtual or real particles over their correlated phase space. The number of particles ranges from few to around 50, described by their energy and three momentum directions, sometimes simplified through on-shell conditions. Typical generative models for the LHC then map simple latent distributions to a phase space distribution encoded in the training data,

$$r \sim p_{\text{latent}}(r) \quad \longleftrightarrow \quad x \sim p_{\text{model}}(x|\theta) \approx p_{\text{data}}(x). \tag{1}$$

The last step represents the network training, for instance in terms of a variational approximation of $p_{\text{data}}(x)$. The latent distribution is typically a standard multi-dimensional Gaussian,

$$p_{\text{latent}}(r) = \mathcal{N}(r; 0, 1). \tag{2}$$

We focus on the case where the dimensionalities of the latent space $r$ and the phase space $x$ are identical, and there is no lower-dimensional latent representation. For these kinds of dimensionalities, bijective network architectures are promising candidates to encode precision correlations. For strictly symmetric bijective networks like INNs the forward and backward directions are inverse to each other, and the network training and evaluation is symmetric. However, this strict symmetry is not necessary to generate LHC events or configurations.

The success of normalizing flows or INNs for this task motivates a study of so-called diffusion or score-based models as an alternative. We will introduce two different models in Sec. 2.1 and 2.2, one with a discrete and one with a continuous time evolution. The main question concerning such diffusion models in LHC physics is if their precision matches the INNs, how we can benefit from their superb expressivity, and if those benefits outweigh the slower evaluation.

A major challenge for all network applications in LHC physics is the required precision in all correlations, and the corresponding power-law scaling with the number of phase space dimensions. This scaling property leads us to introduce and test an autoregressive transformer in Sec. 2.3. Again, the question is how precise and how expressive this alternative approach is and if the benefits justify the extra effort in setup and training.

Because fundamental physics applications require full control and a conservative and reliable uncertainty estimation of neural networks, we will develop Bayesian versions of all three generative models. This allows us to control the uncertainty in the density estimation and to derive an intuition how the different networks learn the phase space distribution of the data.

## 2.1 Denoising diffusion probabilistic model

**Architecture**

Denoising Diffusion Probabilistic Models (DDPM) [73] transform a model density by gradually adding Gaussian noise. This setup guarantees that the network links a non-trivial physics distribution to a Gaussian noise distribution, as illustrated in Eq.(1). The task of the reverse, generative process is to to denoise this diffused data. The structure of diffusion models considers the transformation in Eq.(3) a time-dependent process with $t = 0 \ldots T$,

$$p_{\text{model}}(x_0|\theta) \quad \underset{\leftarrow\text{backward}}{\overset{\text{forward}\rightarrow}{\longleftrightarrow}} \quad p_{\text{latent}}(x_T). \tag{3}$$

The DDPM discretizes the time series in Eq.(3) in the forward direction and encodes is it into a neural network for the backward direction. We start with the forward process, which turns the physical distribution into noise. The corresponding joint distribution is factorized into discrete steps,

$$p(x_1, ..., x_T|x_0) = \prod_{t=1}^{T} p(x_t|x_{t-1}), \qquad \text{with} \qquad p(x_t|x_{t-1}) = \mathcal{N}(x_t; \sqrt{1-\beta_t}x_{t-1}, \beta_t). \tag{4}$$

Each conditional step $p(x_t|x_{t-1})$ adds noise with variance $\beta_t$ around the mean $\sqrt{1-\beta_t}x_{t-1}$. The combination of $x_t$ as a variable and the mean proportional to $x_{t-1}$ implies that the successive steps can be combined as Gaussian convolutions and give the closed form

$$p(x_t|x_0) = \int \prod_{i=1}^{t-1} dx_i \, p(x_t|x_{t-1})p(x_i|x_{i-1})$$

$$= \mathcal{N}(x_t; \sqrt{1-\bar{\beta}_t}x_0, \bar{\beta}_t), \qquad \text{with} \qquad 1-\bar{\beta}_t = \prod_{i=1}^{t}(1-\beta_i). \tag{5}$$

The scaling of the mean with $\sqrt{1-\beta_t}$ prevents the usual addition of the variances and instead stabilizes the evolution of the Gaussian over the time series. The variance can be adapted through a schedule, where $\bar{\beta}_t \rightarrow 1$ for $t \rightarrow T$ should be guaranteed. As suggested in Ref. [73] we choose a linear increase with $\beta_1 = 10^{-7}T$ and $\beta_T = 2 \cdot 10^{-5}T$.

As a first step towards reversing the forward diffusion, we apply Bayes' theorem on each slice defined in Eq.(4),

$$p(x_{t-1}|x_t) = \frac{p(x_t|x_{t-1})p(x_{t-1})}{p(x_t)}. \tag{6}$$

However, a closed-form expression for $p(x_t)$ only exists if conditioned on $x_0$, as given in Eq.(5). Using $p(x_t|x_{t-1}, x_0) = p(x_t|x_{t-1})$ we can instead compute the conditioned forward posterior as a Gaussian

$$p(x_{t-1}|x_t, x_0) = \frac{p(x_t|x_{t-1})p(x_{t-1}|x_0)}{p(x_t|x_0)} = \mathcal{N}(x_{t-1}; \hat{\mu}_t(x_t, x_0), \hat{\beta}_t),$$

$$\text{with} \quad \hat{\mu}(x_t, x_0) = \frac{\sqrt{1-\bar{\beta}_{t-1}}\beta_t}{\bar{\beta}_t}x_0 + \frac{\sqrt{1-\beta_t}\bar{\beta}_{t-1}}{\bar{\beta}_t}x_t, \qquad \text{and} \qquad \hat{\beta}_t = \frac{\bar{\beta}_{t-1}}{\bar{\beta}_t}\beta_t. \tag{7}$$

The actual reverse process starts with Gaussian noise and gradually transforms it into the phase-space distribution through the same discrete steps as Eq.(4), without knowing $x_0$ a

priori. The corresponding generative network needs to approximate Eq.(6) for each step. We start by defining our modeled phase-space distribution

$$p_{\text{model}}(x_0|\theta) = \int dx_1...dx_T \, p(x_0,...,x_T|\theta),$$ (8)

and assume that the joint probability is again given by a chain of independent Gaussians,

$$p(x_0,...,x_T|\theta) = p_{\text{latent}}(x_T)\prod_{t=1}^{T} p_\theta(x_{t-1}|x_t),$$

$$\text{with} \quad p_\theta(x_{t-1}|x_t) = \mathcal{N}(x_{t-1};\mu_\theta(x_t,t),\sigma_\theta^2(x_t,t)).$$ (9)

Here, $\mu_\theta$ and $\sigma_\theta$ are learnable parameters describing the individual conditional probability slices $x_t \rightarrow x_{t-1}$. It turns out that in practice we can fix $\sigma_\theta^2(x_t,t) \rightarrow \sigma_t^2$ [73]. We will see that the advantage of the discrete diffusion model is that we can compare a Gaussian posterior, Eq.(7), with a reverse, learned Gaussian in Eq.(9) for each step.

**Loss function**

Ideally, we want to train our model by maximizing the posterior $p_{\text{model}}(\theta|x_0)$, however, this is not tractable. Using Bayes' theorem and dropping regularization and normalization terms this is equivalent to minimizing the corresponding negative log likelihood in Eqs.(8) and (9),

$$\Big\langle -\log p_{\text{model}}(x_0|\theta) \Big\rangle_{p_{\text{data}}}$$

$$= -\int dx_0 \, p_{\text{data}}(x_0) \log\left(\int dx_1...dx_T \, p_{\text{latent}}(x_T)\prod_{t=1}^{T} p_\theta(x_{t-1}|x_t)\right)$$

$$= -\int dx_0 \, p_{\text{data}}(x_0) \log\left(\int dx_1...dx_T \, p_{\text{latent}}(x_T)p(x_1,...,x_T|x_0)\prod_{t=1}^{T} \frac{p_\theta(x_{t-1}|x_t)}{p(x_t|x_{t-1})}\right)$$

$$= -\int dx_0 \, p_{\text{data}}(x_0) \log\left\langle p_{\text{latent}}(x_T)\prod_{t=1}^{T} \frac{p_\theta(x_{t-1}|x_t)}{p(x_t|x_{t-1})}\right\rangle_{p(x_1,...,x_T|x_0)}.$$ (10)

In the first step, we insert a one into our loss function by dividing Eq.(4) with itself. Using Jensen's inequality $f(\langle x \rangle) \le \langle f(x) \rangle$ for convex functions we find

$$\Big\langle -\log p_{\text{model}}(x_0|\theta) \Big\rangle_{p_{\text{data}}} \le -\int dx_0 \, p_{\text{data}}(x_0) \left\langle \log\left( p_{\text{latent}}(x_T)\prod_{t=1}^{T} \frac{p_\theta(x_{t-1}|x_t)}{p(x_t|x_{t-1})}\right)\right\rangle_{p(x_1,...,x_T)|x_0)}$$

$$= -\left\langle \log\left( p_{\text{latent}}(x_T)\prod_{t=1}^{T} \frac{p_\theta(x_{t-1}|x_t)}{p(x_t|x_{t-1})}\right)\right\rangle_{p(x_0,...,x_T)}$$

$$= \left\langle -\log p_{\text{latent}}(x_T) - \sum_{t=2}^{T} \log \frac{p_\theta(x_{t-1}|x_t)}{p(x_t|x_{t-1})} - \log \frac{p_\theta(x_0|x_1)}{p(x_1|x_0)}\right\rangle_{p(x_0,...,x_T)}$$

$$\equiv \mathcal{L}_{\text{DDPM}}.$$ (11)

As suggested above, we would like to compare each intermediate learned latent distribution $p_\theta(x_{t-1}|x_t)$ to the real posterior distribution $p(x_{t-1}|x_t,x_0)$ of the forward process. To reverse

the ordering of the forward slice we use Bayes' theorem,

$$
\begin{aligned}
\mathcal{L}_{\text{DDPM}} &= \left\langle -\log p_{\text{latent}}(x_T) - \sum_{t=2}^{T} \log \frac{p_\theta(x_{t-1}|x_t)p(x_{t-1}|x_0)}{p(x_{t-1}|x_t,x_0)p(x_t|x_0)} - \log \frac{p_\theta(x_0|x_1)}{p(x_1|x_0)} \right\rangle_{p(x_0,\dots,x_T)} \\
&= \left\langle -\log p_{\text{latent}}(x_T) - \sum_{t=2}^{T} \log \frac{p_\theta(x_{t-1}|x_t)}{p(x_{t-1}|x_t,x_0)} - \log \frac{p(x_1|x_0)}{p(x_T|x_0)} - \log \frac{p_\theta(x_0|x_1)}{p(x_1|x_0)} \right\rangle_{p(x_0,\dots,x_T)} \\
&= \left\langle -\log \frac{p_{\text{latent}}(x_T)}{p(x_T|x_0)} - \sum_{t=2}^{T} \log \frac{p_\theta(x_{t-1}|x_t)}{p(x_{t-1}|x_t,x_0)} - \log p_\theta(x_0|x_1) \right\rangle_{p(x_0,\dots,x_T)} \\
&= \sum_{t=2}^{T} \left\langle \text{KL}[p(x_{t-1}|x_t,x_0), p_\theta(x_{t-1}|x_t)] \right\rangle_{p(x_0,x_t)} + \left\langle -\log p_\theta(x_0|x_1) \right\rangle_{p(x_0,\dots,x_T)} + \text{const.} \\
&\approx \sum_{t=2}^{T} \left\langle \text{KL}[p(x_{t-1}|x_t,x_0), p_\theta(x_{t-1}|x_t)] \right\rangle_{p(x_0,x_t)}.
\end{aligned} \tag{12}
$$

Now, the KL-divergence compares the forward Gaussian step of Eq.(7) with the reverse, learned Gaussian in Eq.(9). The second sampled term will always be negligible compared to the first $T-1$ terms. The KL-divergence between two Gaussian, with means $\mu_\theta(x_t,t)$ and $\hat{\mu}(x_t,x_0)$ and standard deviations $\sigma_t^2$ and $\hat{\beta}_t$, has the compact form

$$
\mathcal{L}_{\text{DDPM}} = \sum_{t=2}^{T} \left\langle \frac{1}{2\sigma_t^2} |\hat{\mu} - \mu_\theta|^2 \right\rangle_{p(x_0,x_t)} + \text{const.} \tag{13}
$$

This implies that $\mu_\theta$ approximates $\hat{\mu}$. The sampling follows $p(x_0,x_t) = p(x_t|x_0)\, p_{\text{data}}(x_0)$. We numerically evaluate this loss using the reparametrization trick on Eq.(5)

$$
\begin{aligned}
x_t(x_0,\epsilon) &= \sqrt{1-\bar{\beta}_t}\, x_0 + \sqrt{\bar{\beta}_t}\,\epsilon\,, \qquad \text{with} \qquad \epsilon \sim \mathcal{N}(0,1) \\
\Leftrightarrow \qquad x_0(x_t,\epsilon) &= \frac{1}{\sqrt{1-\bar{\beta}_t}}\left( x_t - \sqrt{\bar{\beta}_t}\,\epsilon \right).
\end{aligned} \tag{14}
$$

These expressions provide, for example, a closed form for $\hat{\mu}(x_t,x_0)$, but in terms of $x_t$ and $\epsilon$,

$$
\hat{\mu}(x_t,\epsilon) = \frac{1}{\sqrt{1-\beta_t}}\left( x_t(x_0,\epsilon) - \frac{\beta_t}{\sqrt{\bar{\beta}_t}}\epsilon \right). \tag{15}
$$

For the reverse process we choose the same parametrization, but with a learned $\epsilon_\theta(x_t,t)$,

$$
\mu_\theta(x_t,t) \equiv \hat{\mu}(x_t,\epsilon_\theta) = \frac{1}{\sqrt{1-\beta_t}}\left( x_t - \frac{\beta_t}{\sqrt{\bar{\beta}_t}}\epsilon_\theta(x_t,t) \right). \tag{16}
$$

Inserting both expressions into Eq.(13) gives us

$$
\mathcal{L}_{\text{DDPM}} = \sum_{t=2}^{T} \left\langle \frac{1}{2\sigma_t^2} \frac{\beta_t^2}{(1-\beta_t)\bar{\beta}_t} \left| \epsilon - \epsilon_\theta\left( \sqrt{1-\bar{\beta}_t}\, x_0 + \sqrt{\bar{\beta}_t}\,\epsilon, t \right) \right|^2 \right\rangle_{x_0 \sim p_{\text{data}}, \epsilon \sim \mathcal{N}(0,1)}. \tag{17}
$$

The sum over $t$ can be evaluated numerically as a sampling. We chose our model variance $\sigma_t^2 \equiv \hat{\beta}_t$ to follow our true variance. Often, the prefactor in this form for the loss is neglected in the training, but as we need a likelihood loss for the Bayesian setup and no drop in performance was observed, we keep it.

The DDPM model belongs to the broad class of score-based models, and Eq.(13) can also be reformulated for the model to predict the score $s(x_t,t) = \nabla_{x_t} \log p(x_t)$ of our latent space at time $t$. It can be shown that $s_\theta(x_t,t) = -\epsilon_\theta(x_t,t)/\sigma_t$ [82].

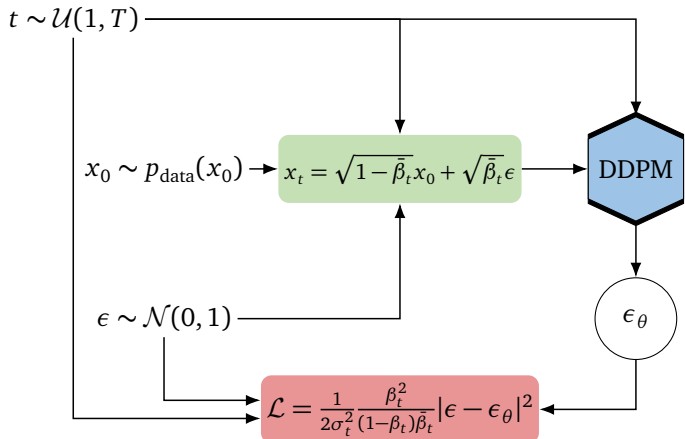

Figure 1: DDPM training algorithm, following Ref. [73], with the loss derived in Eq.(17).

**Training and sampling**

The training algorithm for the DDPM is illustrated in Fig. 1. For a given phase-space point $x_0 \sim p_{\text{data}}(x_0)$ drawn from our true phase space distribution we draw a time step $t \sim \mathcal{U}(1, T)$ from a uniform distribution as well as Gaussian noise $\epsilon \sim \mathcal{N}(0, 1)$ at each iteration. Given Eq.(15) we can then calculate our diffused data after $t$ time steps $x_t$, which is fed to the DDPM network together with our condition $t$. The network encodes $\epsilon_\theta$ and we compare this network prediction with the true Gaussian noise $\epsilon$ multiplied by a $t$-dependent constant as given in the likelihood loss of Eq.(17). Note that we want to ensure that our network sees as many different time steps $t$ for many different phase-space points $x_0$ as necessary to learn the step-wise reversed diffusion process, which is why we use a relatively simple residual dense network architecture, which is trained over many epochs.

The sampling algorithm for the DDPM is shown in Fig. 2. We start by feeding our network our final timestep $T$ and $x_T \sim p_{\text{latent}}(x_T)$ drawn from our Gaussian latent space distribution. With the predicted $\epsilon_\theta$ and drawn Gaussian noise $z_{T-1} \sim \mathcal{N}(0, 1)$ we can then calculate $x_{T-1}$, which is a slightly less diffused version of $x_T$. This procedure is repeated until reaching our phase-space and computing $x_0$, where no additional Gaussian noise is added. Note that during sampling the model needs to predict $\epsilon_\theta$ $T$ times, making the sampling process slower than for classic generative networks like VAEs, GANs, or INNs.

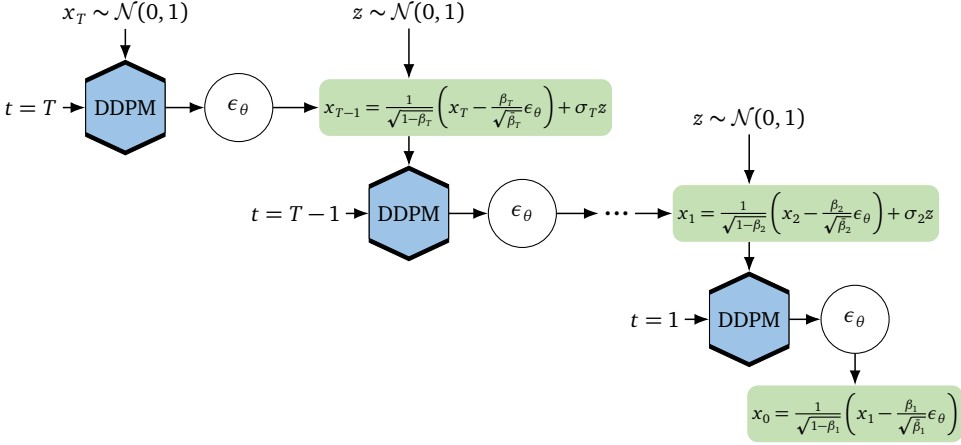

Figure 2: DDPM sampling algorithm, following Ref. [73].

**Likelihood extraction**

To calculate the model likelihood we can use Eq.(8) or its sampled estimate,

$$p_{\text{model}}(x_0|\theta) = \Big\langle p_\theta(x_0|x_1) \Big\rangle_{p(x_1,\dots,x_T|\theta)}, \tag{18}$$

but this is very inefficient. The problem is that $p_\theta(x_0|x_1)$ is a narrow distribution, essentially zero for almost all sampled $x_1$. We can improve the efficiency by importance sampling and use instead

$$
\begin{aligned}
p_{\text{model}}(x_0|\theta) &= \left\langle \frac{p(x_1,\dots,x_T|\theta)}{p(x_1,\dots,x_T|x_0)} p_\theta(x_0|x_1) \right\rangle_{p(x_1,\dots,x_T|x_0)} \\
&= \left\langle \frac{p(x_0,\dots,x_T|\theta)}{p(x_1,\dots,x_T|x_0)} \right\rangle_{p(x_1,\dots,x_T|x_0)}.
\end{aligned} \tag{19}
$$

This samples a diffusion process starting from $x_0$ and into the latent space, meaning that it represents a likely forward and backward path. This means the integrand should not just be zero most of the time.

**Bayesian DDPM**

The key step in the training of generative networks is the density estimation over phase space, from which the network then samples. Like any neural network task this density estimation comes with uncertainties, for instance from a limited amount of training data, a lack of model flexibility, or even training data which we know cannot be trusted. This means that the density estimation step of the generative network should assign an uncertainty to the extracted phase space density, ideally in form of a second map over the target phase space. This problem has been tackled for bijective normalizing flows through a Bayesian network extension [69], which can be combined with other measures, like conditional training on augmented data [22].

The idea behind Bayesian networks is to train network weights as distributions and evaluate the network by sampling over these distributions. This will provide a central value and an uncertainty for the numerically defined network output [63–65].[2] Because general MCMC-methods are expensive for large networks, we use variational inference [83] to learn Gaussian approximations for each weight distribution. Because of the non-linear nature of the network this does not mean that the network output has to come with a Gaussian uncertainty [68].

We repeat the main steps in deriving the Bayesian loss for any neural network approximating, for instance, a density map $\rho(x) \approx \rho_\theta(x)$ following Ref. [84]. The expectation value is defined as

$$\langle \rho \rangle(x) \equiv \langle \rho \rangle = \int d\rho \, \rho \, p(\rho), \qquad \text{with} \qquad p(\rho) = \int d\theta \, p(\rho|\theta) \, p(\theta|x_{\text{train}}), \tag{20}$$

where we omit the $x$-dependence. We use the variational approximation to approximate

$$p(\rho) = \int d\theta \, p(\rho|\theta) \, p(\theta|x_{\text{train}}) \approx \int d\theta \, p(\rho|\theta) \, q(\theta), \tag{21}$$

---

[2]We cannot emphasize often enough that Bayesian networks for uncertainty quantification have nothing to do with Bayesian inference.

where $q(\theta)$ is also a function of $x$. The variational approximation step requires us to minimize

$$
\begin{aligned}
\mathcal{L}_{\text{BNN}} = \text{KL}[q(\theta), p(\theta|x_{\text{train}})] &= \left\langle \log \frac{q(\theta)}{p(\theta|x_{\text{train}})} \right\rangle_q \\
&= \int d\theta \, q(\theta) \, \log \frac{q(\theta)}{p(\theta|x_{\text{train}})} \\
&= - \int d\theta \, q(\theta) \log p(x_{\text{train}}|\theta) + \text{KL}[q(\theta), p(\theta)] + \text{const.}, \quad (22)
\end{aligned}
$$

where we use Bayes' theorem to transform the untractable $p(\theta|x_{\text{train}})$, introducing the prior $p(\theta)$ for the network weights. This so-called ELBO loss combines a likelihood loss with a regularization term, their relative size fixed by Bayes' theorem.

It turns out that for sufficiently deep networks we can choose $q(\theta)$ as uncorrelated Gaussians per network weight [65], such that the training parameters are a set of means and standard deviations for each network weight. Compared to the deterministic network, its Bayesian version is twice the size, but automatically regularized, keeping the additional numerical effort minimal. While $p(\theta)$, also chosen as a Gaussian, is formally defined as a prior, we emphasize that in our case the step from the prior to the posterior has nothing to do with Bayesian inference. The Gaussian width of $p(\theta)$ can be treated as a network hyperparameter and varied to improve the numerical performance. We typically find that the result is stable under varying the width by several orders of magnitude, and width one works well.

The derivation of Eq.(22) can be easily extended to the density estimation step of a generative networks, in the same way as for the Bayesian INN [69]. The Bayesian DDPM loss follows from the deterministic likelihood loss in Eqs.(11) and (17) by adding a sampling over $\theta \sim q(\theta)$ and the regularization term,

$$
\mathcal{L}_{\text{B-DDPM}} = \left\langle \mathcal{L}_{\text{DDPM}} \right\rangle_{\theta \sim q} + \text{KL}[q(\theta), p(\theta)]. \quad (23)
$$

Switching a deterministic network into its Bayesian version includes two steps, (i) swap the deterministic layers to the corresponding Bayesian layers, and (ii) add the regularization term to the loss. For the latter, one complication arises. We estimate the complete loss from a dataset including $N$ events in $M$ batches, which means the likelihood term is summed and then normalized over $M$ batches, while the regularization term comes with the complete prefactor $1/N$.

Table 1: Training setup and hyperparameters for the Bayesian DDPM generator.

| hyperparameter | toy models | LHC events |
|---|---|---|
| Timesteps | 1000 | 1000 |
| Time Embedding Dimension | - | 64 |
| # Blocks | 1 | 2 |
| Layers per Block | 8 | 5 |
| Intermediate Dimensions | 40 | 64 |
| # Model Parameters | 20k | 75k |
| LR Scheduling | one-cycle | one-cycle |
| Starter LR | $10^{-4}$ | $10^{-4}$ |
| Maximum LR | $10^{-3}$ | $10^{-3}$ |
| Epochs | 1000 | 1000, 3000, 10000 |
| Batch Size | 8192 | 8192, 8192, 4096 |
| # Training Events | 600k | 3.2M, 850k, 190k |
| # Generated Events | 1M | 1M, 1M, 1M |

To evaluate the Bayesian network we need to again sample over the network weight distribution. This way we guarantee that the uncertainty of the network output can have any functional form. The number of samplings for the network evaluations can be chosen according to the problem. We choose 30 for all problems discussed in this work. To compare the Bayesian network output with a deterministic network output we can either go into the limit $q(\theta) \to \delta(\theta - \theta_0)$ or only evaluate the means of the network weight distributions.

Our network is implemented in PYTORCH and uses ADAM as optimizer. All hyperparameters are given in Tab. 1. As already mentioned we use a simple residual network which consists of multiple fully connected dense layers with SiLU activation functions. Within our setup a significant increase in performance is achieved when initializing the weights of the last layer in each block to zero.

## 2.2 Conditional flow matching

**Architecture**

As an alternative, we study Conditional Flow Matching (CFM) [74–76]. Like the DDPM, it uses a time evolution to transform phase space samples into noise, so the reverse direction can generate samples as outlined in Eq.(3). Instead of a discrete chain of conditional probabilities, the time evolution of samples in the CFM framework follows a continuous ordinary differential equation (ODE)

$$\frac{dx(t)}{dt} = v(x(t), t), \qquad \text{with} \qquad x(t=0) = x_0, \tag{24}$$

where $v(x(t), t)$ is called the velocity field of the process. This velocity field can be linked to a probability density $p(x, t)$ with the continuity equation

$$\frac{\partial p(x, t)}{\partial t} + \nabla_x [p(x, t) v(x, t)] = 0. \tag{25}$$

These two equations are equivalent in the sense that for a given probability density path $p(x, t)$ any velocity field $v(x, t)$ describing the sample-wise evolution Eq.(24) will be a solution of Eq.(25), and vice versa. Our generative model employs $p(x, t)$ to transforms a phase space distribution into a Gaussian latent distribution

$$p(x, t) \to \begin{cases} p_{\text{data}}(x), & t \to 0, \\ p_{\text{latent}}(x) = \mathcal{N}(x; 0, 1), & t \to 1. \end{cases} \tag{26}$$

The associated velocity field will allow us to generate samples by integrating the ODE of Eq.(24) from $t = 1$ to $t = 0$.

As for the DDPM, we start with a diffusion direction. We define the time evolution from a phase space point $x_0$ to the standard Gaussian as

$$x(t|x_0) = (1-t)x_0 + t\epsilon \to \begin{cases} x_0, & t \to 0, \\ \epsilon \sim \mathcal{N}(0, 1), & t \to 1, \end{cases} \tag{27}$$

following a simple linear trajectory [76], after not finding better results with other choices. For given $x_0$ we can generate $x(t|x_0)$ by sampling

$$p(x, t|x_0) = \mathcal{N}(x; (1-t)x_0, t). \tag{28}$$

This conditional time evolution is similar to the DDPM case in Eq.(5), and it gives us the complete probability path

$$p(x, t) = \int dx_0\, p(x, t|x_0)\, p_{\text{data}}(x_0). \tag{29}$$

It fulfills the boundary conditions in Eq.(26),

$$p(x,0) = \int dx_0 \, p(x,0|x_0) \, p_{\text{data}}(x_0) = \int dx_0 \, \delta(x-x_0) \, p_{\text{data}}(x_0) = p_{\text{data}}(x),$$

$$p(x,1) = \int dx_0 \, p(x,1|x_0) \, p_{\text{data}}(x_0) = \mathcal{N}(x;0,1) \int dx_0 \, p_{\text{data}}(x_0) = \mathcal{N}(x;0,1). \quad (30)$$

From this probability density path we need to extract the velocity field. We start with the conditional velocity, associated with $p(x,t|x_0)$, and combine Eq.(24) and (27) to

$$v(x(t|x_0),t|x_0) = \frac{d}{dt}[(1-t)x_0 + t\epsilon] = -x_0 + \epsilon. \quad (31)$$

The linear trajectory leads to a time-constant velocity, which solves the continuity equation for $p(x,t|x_0)$ by construction. We exploit this fact to find the unconditional $v(x,t)$

$$\frac{\partial p(x,t)}{\partial t} = \int dx_0 \, \frac{\partial p(x,t|x_0)}{\partial t} \, p_{\text{data}}(x_0)$$

$$= -\int dx_0 \, \nabla_x \left[ v(x,t|x_0) p(x,t|x_0) \right] p_{\text{data}}(x_0)$$

$$= -\nabla_x \left[ p(x,t) \int dx_0 \, \frac{v(x,t|x_0) p(x,t|x_0) p_{\text{data}}(x_0)}{p(x,t)} \right]$$

$$= -\nabla_x \left[ p(x,t) v(x,t) \right], \quad (32)$$

by defining

$$v(x,t) = \int dx_0 \, \frac{v(x,t|x_0) p(x,t|x_0) p_{\text{data}}(x_0)}{p(x,t)}. \quad (33)$$

While the conditional velocity in Eq.(31) describes a trajectory between a normal distributed and a phase space sample $x_0$ that is specified in advance, the aggregated velocity in Eq.(33) can evolve samples from $p_{\text{data}}$ to $p_{\text{latent}}$ and vice versa.

Like the DDPM model, the CFM model can be linked to score-based diffusion models, [74] derive a general relation between the velocity field and the score of a diffusion process that for our linear trajectory reduces to $s(x,t) = -\frac{1}{t}(x + (1-t)v(x,t))$.

**Loss function**

Encoding the velocity field in Eq.(33) is a simple regression task, $v(x,t) \approx v_\theta(x,t)$. The straightforward choice for the loss is the mean squared error,

$$\left\langle [v_\theta(x,t) - v(x,t)]^2 \right\rangle_{t,x\sim p(x,t)} = \left\langle v_\theta(x,t)^2 \right\rangle_{t,x\sim p(x,t)} - \left\langle 2v_\theta(x,t)v(x,t) \right\rangle_{t,x\sim p(x,t)} + \text{const.}, \quad (34)$$

where the time is sampled uniformly over $t \in [0,1]$. While we would want to sample $x$ from the probability path given in Eq.(29) and learn the velocity field given in Eq.(33), neither of those is tractable. However, it would be easy to sample from the conditional path in Eq.(28) and calculate the conditional velocity in Eq.(31). We rewrite the above loss in terms of the conditional quantities, so the first term becomes

$$\left\langle v_\theta(x,t)^2 \right\rangle_{t,x\sim p(x,t)} = \left\langle \int dx \, v_\theta(x,t)^2 \int dx_0 \, p(x,t|x_0) p_{\text{data}}(x_0) \right\rangle_t$$

$$= \left\langle v_\theta(x,t)^2 \right\rangle_{t,x_0\sim p_{\text{data}}, x\sim p(x,t|x_0)}$$

$$= \left\langle v_\theta(x(t|x_0),t)^2 \right\rangle_{t,x_0\sim p_{\text{data}},\epsilon}. \quad (35)$$

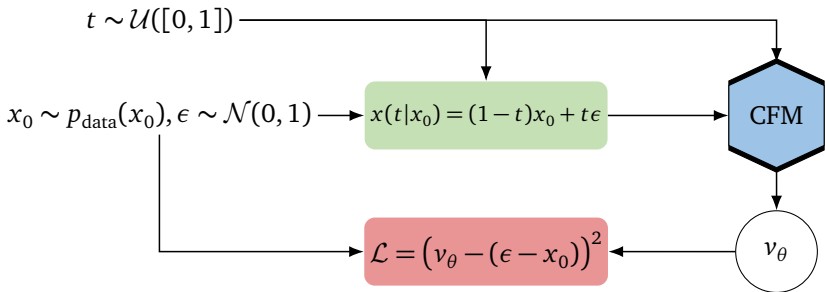

Figure 3: CFM training algorithm, with the loss derived in Eq.(37).

Using Eq.(33) we can rewrite the second loss term as

$$
\begin{aligned}
-2\Big\langle v_\theta(x,t)v(x,t)\Big\rangle_{t,x\sim p(x,t)} &= -2\Bigg\langle \int dx\, p(x,t)v_\theta(x,t)\,\frac{\int dx_0 v(x,t|x_0)p(x,t|x_0)p_{\text{data}}(x_0)}{p(x,t)}\Bigg\rangle_t \\
&= -2\Bigg\langle \int dx dx_0\, v_\theta(x,t)\, v(x,t|x_0)\, p(x,t|x_0)\, p_{\text{data}}(x_0)\Bigg\rangle_t \\
&= -2\Big\langle v_\theta(x,t)\, v(x,t|x_0)\Big\rangle_{t,x_0\sim p_{\text{data}},x\sim p(x,t|x_0)} \\
&= -2\Big\langle v_\theta(x(t|x_0),t)\, v(x(t|x_0),t|x_0)\Big\rangle_{t,x_0\sim p_{\text{data}},\epsilon}\,.
\end{aligned}
\tag{36}
$$

The (conditional) Flow Matching loss of Eq.(34) then becomes

$$
\begin{aligned}
\mathcal{L}_{\text{CFM}} &= \Big\langle [v_\theta(x(t|x_0),t)-v(x(t|x_0),t|x_0)]^2\Big\rangle_{t,x_0\sim p_{\text{data}},\epsilon} \\
&= \Bigg\langle \bigg[v_\theta(x(t|x_0),t)-\frac{dx(t|x_0))}{dt}\bigg]^2\Bigg\rangle_{t,x_0\sim p_{\text{data}},\epsilon} \\
&= \Big\langle [v_\theta((1-t)x_0+t\epsilon,t)-(\epsilon-x_0)]^2\Big\rangle_{t,x_0\sim p_{\text{data}},\epsilon}\,.
\end{aligned}
\tag{37}
$$

**Training and sampling**

The CFM training is illustrated in Fig. 3. At each iteration we sample a data point $x_0 \sim p_{\text{data}}(x_0)$ and a normal distributed $\epsilon \sim \mathcal{N}(0,1)$ as starting and end points of a trajectory, as well as a time $t \sim \mathcal{U}([0,1])$. We then compute $x(t|x_0)$ following Eq.(27) and the associated conditional velocity $v(x(t|x_0),t|x_0)$ following Eq.(31). The point $x(t|x_0)$ and the time $t$ are passed to a neural network which encodes the conditional velocity field $v_\theta(x,t) \approx v(x,t|x_0)$. One property of the training algorithm is that the same network input, a time $t$ and a position $x(t|x_0)$, can be produced by many different trajectories, each with a different conditional velocity. While the network training is based on a wide range of possible trajectories, the CFM loss in Eq.(37) ensures that sampling over many trajectories returns a well-defined velocity field.

Once the CFM model is trained, the generation of new samples is straightforward. We start by drawing a sample from the latent distribution $x_1 \sim p_{\text{latent}} = \mathcal{N}(0,1)$ and calculate its time evolution by numerically solving the ODE backwards in time from $t=1$ to $t=0$

$$
\frac{d}{dt}x(t) = v_\theta(x(t),t), \qquad \text{with} \qquad x_1 = x(t=1) \quad \Rightarrow \quad x_0 = x_1 - \int_0^1 v_\theta(x,t)dt \equiv G_\theta(x_1). \tag{38}
$$

We use the `scipy.solve_ivp` function with default settings for this. Under mild regularity assumptions this solution defines a bijective transformation between the latent space sample and the phase space sample $G_\theta(x_1)$, similar to an INN.

Table 2: Training setup and hyperparameters for the Bayesian CFM generator.

| hyperparameter | toy models | LHC events |
|---|---|---|
| Embedding Dimension | - | 32 |
| # Blocks | 1 | 2 |
| Layers per Block | 8 | 5 |
| Intermediate Dimensions | 40 | 128, 64, 64 |
| # Model Parameters | 20k | 265k, 85k, 85k |
| LR Scheduling | cosine annealing | cosine annealing |
| Starter LR | $10^{-2}$ | $10^{-3}$ |
| Epochs | 1000 | 1000, 5000, 10000 |
| Batch Size | 8192 | 16384 |
| # Training Events | 600k | 3.2M, 850k, 190k |
| # Generated Events | 1M | 1M, 1M, 1M |

**Likelihood extraction**

The CFM model also allows to calculate phase space likelihoods. Making use of the continuity equation we can write

$$
\begin{aligned}
\frac{dp(x,t)}{dt} &= \frac{\partial p(x,t)}{\partial t} + \nabla_x p(x,t)\, v(x,t) \\
&= \frac{\partial p(x,t)}{\partial t} + \nabla_x [p(x,t)v(x,t)] - p(x,t)\nabla_x v(x,t) \\
&= -p(x,t)\nabla_x v(x,t).
\end{aligned}
\tag{39}
$$

Its solution is

$$
\frac{p(x_1,1)}{p(x_0,0)} \equiv \frac{p_{\text{latent}}(G_\theta^{-1}(x_0))}{p_{\text{model}}(x_0|\theta)} = \exp\left(-\int_0^1 dt\, \nabla_x v(x(t),t)\right),
\tag{40}
$$

and we can write in the usual INN notation [84]

$$
p_{\text{model}}(x_0|\theta) = p_{\text{latent}}(G_\theta^{-1}(x_0)) \left| \det \frac{\partial G_\theta^{-1}(x_0)}{\partial x_0} \right|
$$

$$
\Rightarrow \quad \left| \det \frac{\partial G_\theta^{-1}(x_0)}{\partial x_0} \right| = \exp\left(\int_0^1 dt\, \nabla_x v_\theta(x(t),t)\right).
\tag{41}
$$

Calculating the Jacobian requires integrating over the divergence of the learned velocity field. This divergence can be calculated using automatic differentiation approximately as fast as $n$ network calls, where $n$ is the data dimensionality.

**Bayesian CFM**

Finally, we also turn the CFM into a Bayesian generative model, to account for the uncertainties in the underlying density estimation [69]. From the Bayesian DDPM we know that this can be achieved by promoting the network weights from deterministic values to, for instance, Gaussian distributions and using variational approximation for the training [63–65, 83]. For the Bayesian INN or the Bayesian DDPM the loss is a sum of the likelihood loss and a KL-divergence regularization, Eq.(23). Unfortunately, the CFM loss in Eq.(37) is not a likelihood

loss. To construct a Bayesian CFM loss we therefore combine it with Bayesian network layers and a free KL-regularization,

$$\mathcal{L}_{\text{B-CFM}} = \left\langle \mathcal{L}_{\text{CFM}} \right\rangle_{\theta \sim q(\theta)} + c \, \text{KL}[q(\theta), p(\theta)]. \tag{42}$$

While for a likelihood loss the factor $c$ is fixed by Bayes' theorem, in the CFM case it is a free hyperparameter. We find that the network predictions and their associated uncertainties are very stable when varying it over several orders of magnitude.

Our network is implemented in PYTORCH and uses ADAM as optimizer. All hyperparameters are given in Tab. 2. We employ a simple network consisting of fully connected dense layers with SiLU activation functions. Given limited resources, simple and fast networks trained for a large number of iterations produces the best results. For the LHC events we used two blocks of dense layers connected by a residual connection. In our setup dropout layers lead to significantly worse results, while normalization layers have no visible impact on the results. We find that the training of CFM models can be very noisy, using a large batch size can help to stabilize this.

In general, training diffusion models requires a relatively large number of epochs, as indicated in Tabs. 1 and 2. A key result of our study is to use a cosine-annealing learning rate scheduler for the CFMs and one-cycle scheduling for the DDPM, as well as significantly downsizing the models compared to INNs, to allow for more training epochs. For the entire hyperparameter setup, our B-DDPM implementation turns out to be slightly more sensitive than the B-CFM.

### 2.3 Autoregressive transformer

**Architecture**

A distinct shortcoming of traditional generative models like GANs, INNs, and diffusion models is that they learn the correlations in all phase space directions simultaneously. This leads to a power-law scaling for instance of the training effort for a constant precision in the learned correlations [77]. The autoregressive transformer (AT) [85] instead interprets the phase space vector $x = (x_1, ...x_n)$ as a sequence of elements $x_i$ and factorizes the joint $n$-dimensional probability into $n$ probabilities with a subset of conditions,

$$p_{\text{model}}(x|\theta) = \prod_{i=1}^{n} p(x_i|x_1, ..., x_{i-1}) \approx p_{\text{data}}(x), \tag{43}$$

as illustrated in Fig. 4. This autoregressive approach improves the scaling with the phase space dimensionality in two ways. First, each distribution $p(x_i|x_1, ...x_{i-1})$ is easier to learn than a distribution conditional on the full phase space vector $x$. Second, we can use our physics knowledge to group challenging phase space directions early in the sequence $x_1, ..., x_n$.

The network learns the conditional probabilities over phase space using a representation

$$p(x_i|\omega^{(i-1)}) = p(x_i|x_1, ...x_{i-1}), \tag{44}$$

where the parameters $\omega^{(i-1)}$ encode the conditional dependence on $x_1, ...x_{i-1}$. A naive choice are binned probabilities $w_j^{(i-1)}$ per phase space direction,

$$p(x_i|\omega^{(i-1)}) = \sum_{\text{bins } j} w_j^{(i-1)} \mathbb{1}^{(j)}(x_i), \tag{45}$$

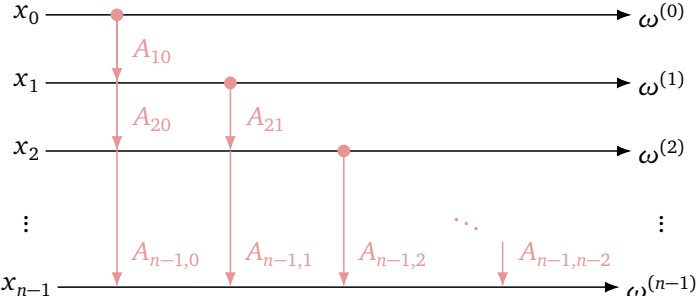

Figure 4: Autoregressive approach to density estimation. The attention matrix $A_{ij}$ defined in Eq.(50) encodes information between components $x_i$. We introduce an auxiliary condition $x_0 = 0$ for the first phase space component $x_1$.

where $\mathbb{1}^{(j)}(x)$ is one for $x$ inside the bin $j$ and zero outside. A more flexible and better-scaling approach is a Gaussian mixture,

$$p(x_i|\omega^{(i-1)}) = \sum_{\text{Gaussian } j} w_j^{(i-1)} \mathcal{N}(x_i; \mu_j^{(i-1)}, \sigma_j^{(i-1)}). \tag{46}$$

It generalizes the fixed bins to a set of learnable means and widths.

Our architecture closely follows the Generative Pretrained Transformer (GPT) models [85], illustrated in Fig. 5. The network takes a sequence of $x_i$ as input and evaluates them all in parallel. We use a linear layer to map each value $x_i$ in a $d$-dimensional latent space, denoted as $x_{i\alpha}$. The network consists of a series of TransformerDecoder blocks, combining a self-attention layer with a standard feed-forward network. Finally, a linear layer maps the latent space onto the representation $\omega^{(i-1)}$ of the conditions.

Equations (45) and (46) do not provide an actual structure correlating phase space regions and phase space directions. This means the transformer needs to construct an appropriate basis and correlation pattern by transforming the input $x$ into an $x'$, with the same dimension as the input vector and leading to the $\omega$ representation. Its goal is to construct a matrix $A_{ij}$ that quantifies the relation or similarity of two embedded phase space components $x_{i\alpha}$ and $x_{j\alpha}$. We construct the single-headed self-attention [86] of an input $x$ in three steps.

1. Using the conventions of the first layer, we want to measure the relation between $x_i$ and a given $x_j$, embedded in the $d$-dimensional latent space. Replacing the naive scalar product $x_{i\alpha}x_{j\alpha}$, we introduce learnable latent-space transformations $W^{Q,K}$ to the elements

$$q_{i\alpha} = W_{\alpha\beta}^Q x_{i\beta}, \qquad \text{and} \qquad k_{j\alpha} = W_{\alpha\beta}^K x_{j\beta}, \tag{47}$$

and use the directed scalar product,

$$A_{ij} \sim q_{i\alpha}k_{j\alpha}, \tag{48}$$

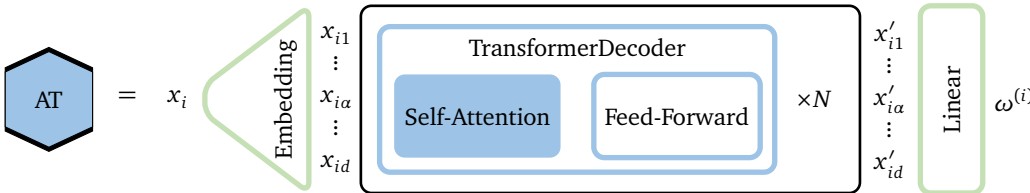

Figure 5: Architecture of the autoregressive transformer. All phase space components $x_i$ are evaluated in parallel, see Fig. 4.

to encode the relation of $x_j$ with $x_i$ through $k_j$ and $q_i$. While the scalar product is symmetric, the attention matrix does not have to be, $A_{ij} \neq A_{ji}$. These global transformations allow the transformer to choose a useful basis for the scalar product in latent space.

2. The first problem with $A_{ij}$ given in Eq.(48) is that it grows with the latent space dimensionality, so it turns out to be useful to replace it by $A_{ij} \rightarrow A_{ij}/\sqrt{d}$. More importantly, we want all entries $j$ referring to a given $i$ to be normalized,

$$A_{ij} \in [0, 1], \qquad \text{and} \qquad \sum_j A_{ij} = 1. \tag{49}$$

This leads us to the definition

$$A_{ij} = \text{Softmax}_j \frac{q_{i\alpha} k_{j\alpha}}{\sqrt{d}}, \qquad \text{with} \qquad \text{Softmax}_j(x_j) = \frac{e^{x_j}}{\sum_k e^{x_k}}. \tag{50}$$

Similar to the adjacency matrix of a graph, this attention matrix quantifies how closely two phase space components are related. Our autoregressive setup sketched in Fig. 4 requires us to set

$$A_{ij} = 0, \qquad \text{for} \qquad j > i. \tag{51}$$

3. Now that the network has constructed a basis to evaluate the relation between two input elements $x_i$ and $x_j$, we use it to update the actual representation of the input information. We combine the attention matrix $A_{ij}$ with the input data, but again transformed in latent space through a learnable matrix $W^V$,

$$v_{j\alpha} = W^V_{\alpha\beta} x_{j\beta} \quad \Rightarrow \quad x'_{i\alpha} = A_{ij} v_{j\alpha}$$
$$= \text{Softmax}_j \left( \frac{W^Q_{\delta\gamma} x_{i\gamma} W^K_{\delta\sigma} x_{j\sigma}}{\sqrt{d}} \right) W^V_{\alpha\beta} x_{j\beta}. \tag{52}$$

In this form we see that the self-attention vector $x'$ just follows from a general basis transformation with the usual scalar product, but with an additional learned transformation for every input vector.

The self-attention can be stacked with other structures like a feed-forward network, to iteratively construct an optimal latent space representation. This can either be identified with the final output $\omega^{(i)}$ or linked to this output through a simple linear layer. To guarantee a stable training of this complex structure, we evaluate the self-attention as an ensemble, defining a multi-headed self-attention. In addition, we include residual connections, layer normalization, and dropout just like the GPT model. Because the sum over $j$ in Eq.(52) leads to permutation equivariance in the phase space components, we break it by providing explicit positional information through a linear layer that takes the one-hot encoded phase space position $i$ as input. This positional embedding is then added to the latent representation $x_{i\alpha}$.

**Training and sampling**

The training of the autoregressive transformer is illustrated in Fig. 6. We start with an universal $x_0 = 0$ in $p(x_1|\omega^{(0)})$ for all events. The transformer encodes all parameters $\omega$ needed for $p(x_i|\omega^{(i-1)})$ in parallel. The chain of conditional likelihoods for the realized values $x_i$ gives the full likelihood $p_{\text{model}}(x|\theta)$, which in turn can be used for the loss function

$$\mathcal{L}_{\text{AT}} = \left\langle -\log p_{\text{model}}(x|\theta) \right\rangle_{x \sim p_{\text{data}}}$$
$$= \sum_{i=1}^n \left\langle -\log p(x_i|\omega^{(i-1)}) \right\rangle_{x \sim p_{\text{data}}}. \tag{53}$$

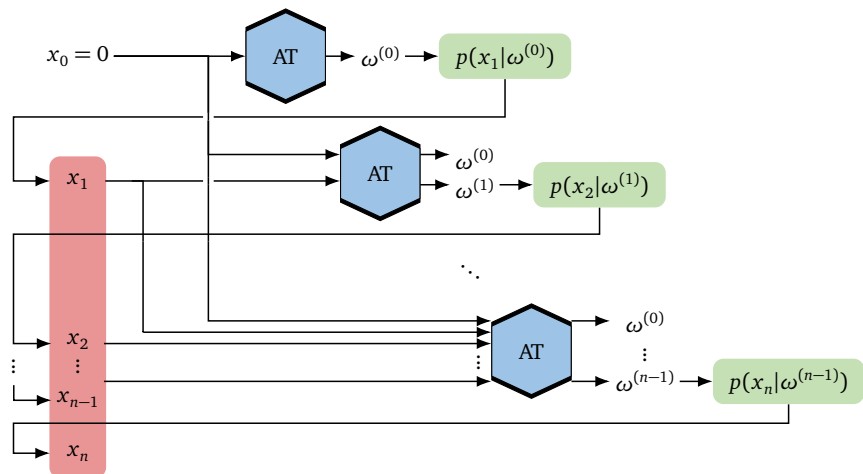

Figure 6: Training algorithm for the autoregressive transformer.

The successive transformer sampling is illustrated in Fig. 7. For each component, $\omega^{(i-1)}$ encodes the dependence on the previous components $x_1, ..., x_{i-1}$, and correspondingly we sample from $p(x_i|\omega^{(i-1)})$. The parameters $\omega^{(0)}, ... \omega^{(i-2)}$ from the sampling of previous components are re-generated in each step, but not used further. This way the event generation is less efficient than the likelihood evaluation during training, because it cannot be parallelized.

**Bayesian version**

As any generative network, we bayesianize the transformer by drawing its weights from a set of Gaussians $q(\theta)$ as defined Eq.(21). In practice, we replace the deterministic layers of the transformer by Bayesian layers and add the KL-regularization from Eq.(22) to the likelihood loss of the transformer, Eq.(53)

$$\mathcal{L}_{\text{B-AT}} = \left\langle \mathcal{L}_{\text{AT}} \right\rangle_{\theta \sim q(\theta)} + \text{KL}[q(\theta), p(\theta)]. \tag{54}$$

For large generative networks, we encounter the problem that too many Bayesian weights destabilize the network training. While a deterministic network can switch of unused weights by just setting them to zero, a Bayesian network can only set the mean to zero, in which case the Gaussian width will approach the prior $p(\theta)$. This way, excess weights can contribute noise to the training of large networks. This problem can be solved by adjusting the hyperparameter describing the network prior or by only bayesianizing a fraction of the network weights. In both cases it is crucial to confirm that the uncertainty estimate from the network is on a stable plateau. For the transformer we find that the best setup is to only bayesianizing the last layer.

To implement the autoregressive transformer we use PYTORCH with the RADAM optimizer. All hyperparameters are given in Tab. 3. We propose to couple the number of parameters $m$ in the parametrization vector $\omega^{(i-1)}$ to the latent space dimensionality $d$, because the latent space dimensionality naturally sets the order of magnitude of parameters that the model can predict confidently.

Figure 7: Sampling algorithm for the autoregressive transformer.

Table 3: Training setup and hyperparameters for the Bayesian autoregressive transformer.

| hyperparameter | toy models | LHC events |
|---|---|---|
| # Gaussians $m$ | 21 | 43 |
| # Bins $m$ | 64 | - |
| # TransformerDecoder $N$ | 4 | 4 |
| # Self-attention Heads | 4 | 4 |
| Latent Space Size $d$ | 64 | 128 |
| # Model Parameters | 220k | 900k |
| LR Scheduling | one-cycle | one-cycle |
| Starter LR | $3 \times 10^{-4}$ | $10^{-4}$ |
| Maximum LR | $3 \times 10^{-3}$ | $10^{-3}$ |
| Epochs | 200 | 2000 |
| Batch Size | 1024 | 1024 |
| RADAM $\epsilon$ | $10^{-8}$ | $10^{-4}$ |
| # Training Events | 600k | 2.4M, 670k, 190k |
| # Generated Events | 600k | 1M, 1M, 1M |

## 3  Toy models and Bayesian networks

Before we can turn to the LHC phase space as an application to our novel generative models, we study their behavior for two simple toy models, directly comparable to Bayesian INNs [69]. These toy models serve two purposes: first, we learn about the strengths and the challenges of the different network architectures, when the density estimation task is simple and the focus lies on precision. Second, the interplay between the estimation of the density and its uncertainty over phase space allows us to understand how the different network encode the density. We remind ourselves that an INN just works like a high-dimensional fit to the correlated 2-dimensional densities [69].

**Denoising diffusion probabilistic model**

Our first toy example is a normalized ramp, linear in one direction and flat in the second,

$$p_{\mathrm{ramp}}(x_1, x_2) = 2x_2 \,. \tag{55}$$

The network input and output are unweighted events. The hyperparameters of each model are given in Tabs. 1, 2, and 3. A training dataset of 600k events guarantees that for our setup and binning the statistical uncertainty on the phase space density is around the per-cent level. To show one-dimensional Bayesian network distributions we sample the $x_i$-direction and the $\theta$-space in parallel [22, 69]. This way the uncertainty in one dimension is independent of the existence and size of other dimensions.

Starting with the DDPM we show the non-trivial one-dimensional distributions in Fig. 8. In the left panel we see that the network learns the underlying phase space density well, but not quite at the desired per-cent precision. The uncertainty from the B-DDPM captures remaining deviations, if anything, conservatively. In the right panel we see that the absolute uncertainty has a minimum around $x_1 = 0.7$, similar to the behavior of the Bayesian INN and confirmed by independent trainings. We can understand this pattern by looking at a constrained fit of the normalized density

$$p(x_2) = a x_2 + b = a \left( x_2 - \frac{1}{2} \right) + 1 \,, \qquad \text{with} \qquad x_2 \in [0, 1] \,. \tag{56}$$

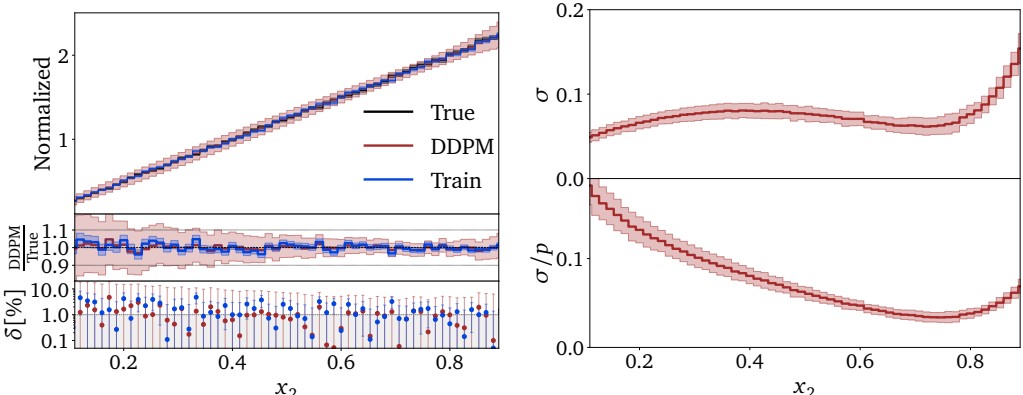

Figure 8: Ramp distribution from the DDPM. We show the learned density and its B-DDPM uncertainty (left) as well as the absolute and relative uncertainties with a range given by 10 independent trainings (right). We use $\delta = |\text{Model} - \text{Truth}|/\text{Truth}$.

A fit of $a$ then leads to an uncertainty in the density of

$$\sigma \equiv \Delta p \approx \left| x_2 - \frac{1}{2} \right| \Delta a, \tag{57}$$

just using simple error propagation. The minimum in the center of the phase space plan can be interpreted as the optimal use of correlations in all directions to determine the local density.

For the DDPM the minimum is not quite at $x_2 = 0.5$, and the uncertainty as a function of $x_2$ is relatively flat over the entire range. Because of the statistically limited training sample, the network output comes with a relatively large uncertainty towards $x_2 = 0$. For larger $x_2$-values, the gain in precision and uncertainty is moderate. For $x_2 > 0.75$ the absolute and relative uncertainties increase, reflecting the challenge to learn the edge at $x_2 = 1$. These results are qualitatively similar, but quantitatively different from the INN case, which benefits more from the increase in training data and correlations for $x_2 = 0.1 \dots 0.5$.

The second toy example is a Gaussian ring, or a Gaussian sphere in two dimensions,

$$p_{\text{ring}}(x_1, x_2) = \mathcal{N}(\sqrt{x_1^2 + x_2^2}; 1, 0.1). \tag{58}$$

The DDPM result are shown in Fig. 9. The precision on the density is significantly worse than for the ramp, clearly missing the per-cent mark. The agreement between the training data

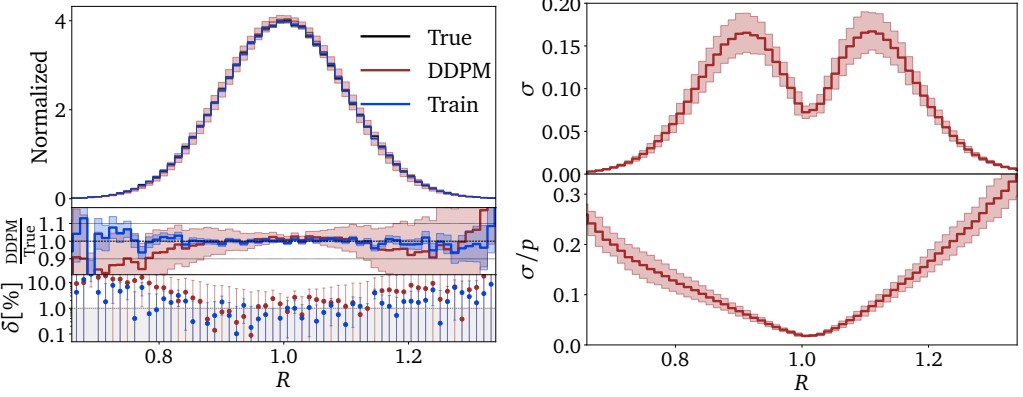

Figure 9: Gaussian ring distribution from the DDPM. We show the learned density and its B-DDPM uncertainty (left) as well as the absolute and relative uncertainties with a range given by 10 independent trainings (right).

and the learned density is not quite symmetric, reflecting the fact that we train and evaluate the network in Cartesian candidates but show the result in $R$. Especially for large radii, the network significantly overestimates the tail, a failure mode which is covered by the predictive uncertainty only for $R \lesssim 1.3$. In the right panels of Fig. 9 the main feature is a distinct minimum in the uncertainty around the mean of the Gaussian. As for the ramp, this can be understood from error propagation in a constrained fit. If we assume that the network first determines a family of functions describing the radial dependence, in terms of a mean and a width, the contribution from the mean vanishes at $R = 1$ [69]. Alternatively, we can understand the high confidence of the network through the availability of many radial and angular correlations in this phase space region.

**Conditional flow matching**

To confirm that the diffusion architecture is behind the DDPM features, we repeat our study with the CFM model in Fig. 10. The main difference to the DDPM is that the agreement between the learned and the training densities is now at the per-cent level, for the ramp and for the Gaussian ring. This shows that diffusion models are indeed able to learn a phase space density with the same precision and stability as normalizing flows or INNs. As before, the predictive uncertainty from the B-CFM model is conservative for the entire phase space of the ramp, but it fails in the exponentially suppressed tail of the Gaussian ring for $R \gtrsim 1.3$. We emphasize that as a function of $R$ this problem is clearly visible when we increase $R$ to the point where $\sigma(R) = \mathcal{O}(p(R))$.

Looking at the pattern of the predicted uncertainty $\sigma$ in $x_2$ and in $R$, we see a similar

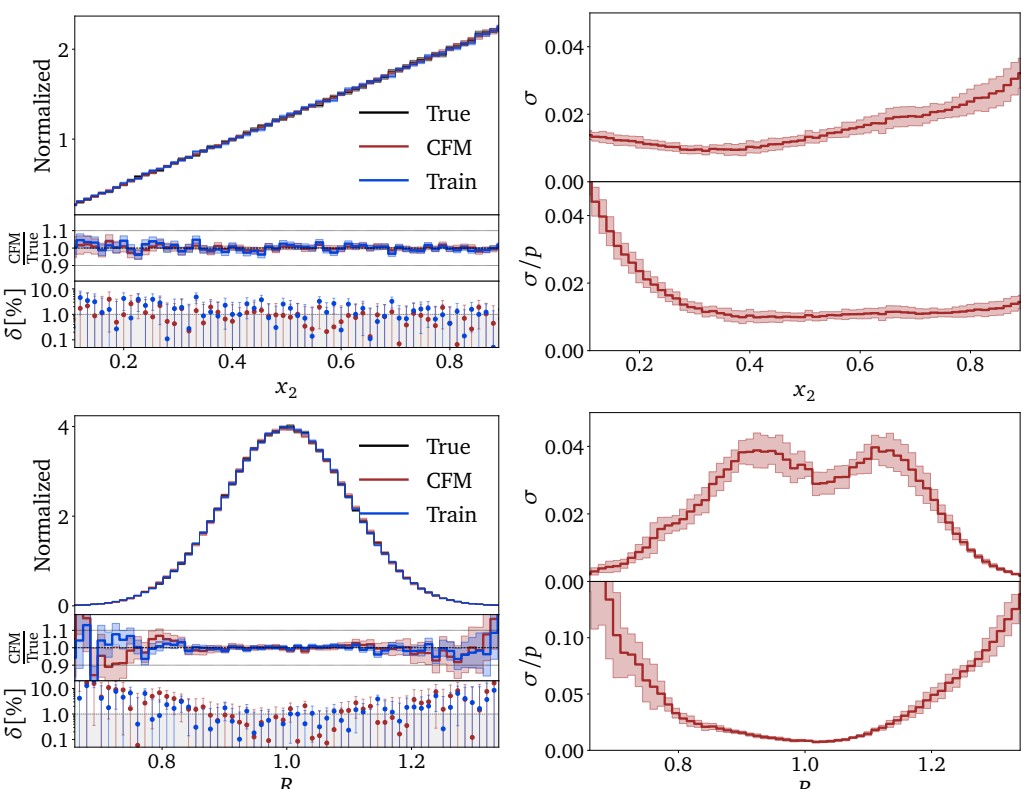

Figure 10: Ramp (upper) and Gaussian ring (lower) distributions from the CFM. We show the learned density and its B-CFM uncertainty (left) as well as the absolute and relative uncertainties with a range given by 10 independent trainings (right).

behavior as for the INN and for the DDPM. As for the DDPM, the minimum in the middle of the ramp is flatter than for the INN, and its position has moved to $x_2 \approx 0.3$. For the radial distribution of the Gaussian ring there is the usual minimum on the peak.

Summarizing our findings for the two diffusion models, they behave similar but not identical to the INN. For all of them, the relation between the density and its uncertainty shows patterns of a constrained fit, suggesting that during the the training the networks first determine a class of suitable models and then adjust the main features of these models, like the slope of a ramp or the position and width of a Gaussian ring.

**Autoregressive transformer**

Finally, we target the two-dimensional ramp, Eq.(55), and the Gaussian ring, Eq.(58) with the transformer. In Fig. 11 we start with a simple representation of the phase space density using 64 bins. In this naive setup the densities of the ramp and the Gaussian ring are described accurately, within our per-cent target range. The largest deviations appear in the tails of the Gaussian ring, but remain almost within the statistical limitations of the training data.

Unlike for the INN and the diffusion models, the uncertainty in the right panels of Fig. 11 does not show any real features for the ramp or the Gaussian ring. This shows that the transformer does not use a fit-like density estimation and does not benefit from the increased correlations in the center of phase space. Both of these aspects can be understood from the model setup. First, the autoregressive structure never allows the transformer to see the full phase

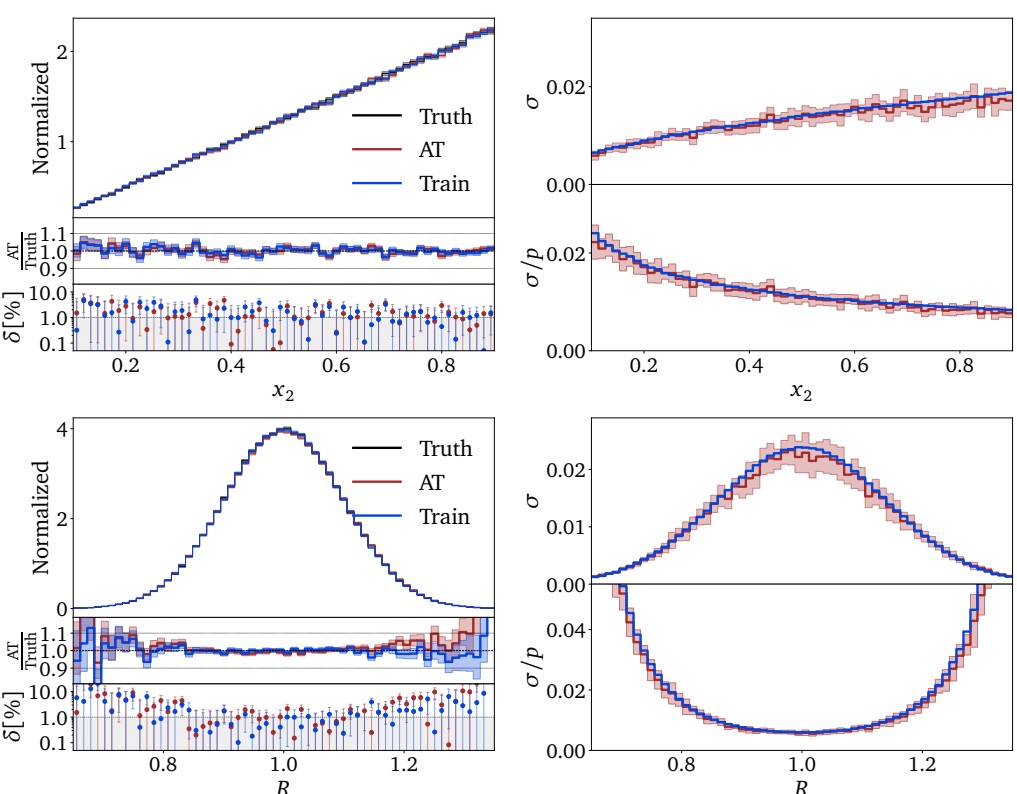

Figure 11: Ramp (upper) and Gaussian ring (lower) distribution from the autoregressive transformer with a binned likelihood. We show the learned density and its Bayesian network uncertainty (left) as well as the absolute and relative uncertainties with a range given by 10 independent trainings, compared to the statistical uncertainty of the training data in blue (right).

space density and encode global (symmetry) patterns; second, the main motivation of the transformer is to improve the power-law scaling with the dimensionality of all possible correlations and only focus on the most relevant correlations at the expense of the full phase space coverage.

In Fig. 12 we show the same results for a mixture of 21 Gaussians. For this small number of dimensions the advantage over the binned distribution is not obvious. The main problem appears at the upper end of the ramp, where there exists enough training data to determine a well-suited model, but the poorly-suited GMM just fails to reproduce the flat growth towards the sharp upper edge and introduces a significant artifact, just covered by the uncertainty. For the Gaussian ring the GMM-based transformer is also less precise than the binned version, consistent with the lower resolution in the 2-dimensional model.

The uncertainty predicted by the Bayesian transformer is typically smaller than for diffusion models. We therefore add the statistical uncertainty of the training data to the right panels of Figs. 11 and 12, providing a lower bound on the uncertainty. In both cases, the uncertainty of the Bayesian transformer conservatively tracks the statistical uncertainty of the training data.

Finally, in Fig. 13 we illustrate the unique way in which the GMM-based transformer reconstructs the density for the Gaussian ring successively. In the left panel, we show $p_{\text{model}}(x_1)$ after the first autoregressive step, constructed out of 21 learned Gaussians. The peaks at $\pm 1$ arise from the marginalization along the longest line of sight. The marginalization also distorts the form of the Gaussians, which are distributed along the ring. The density after the second

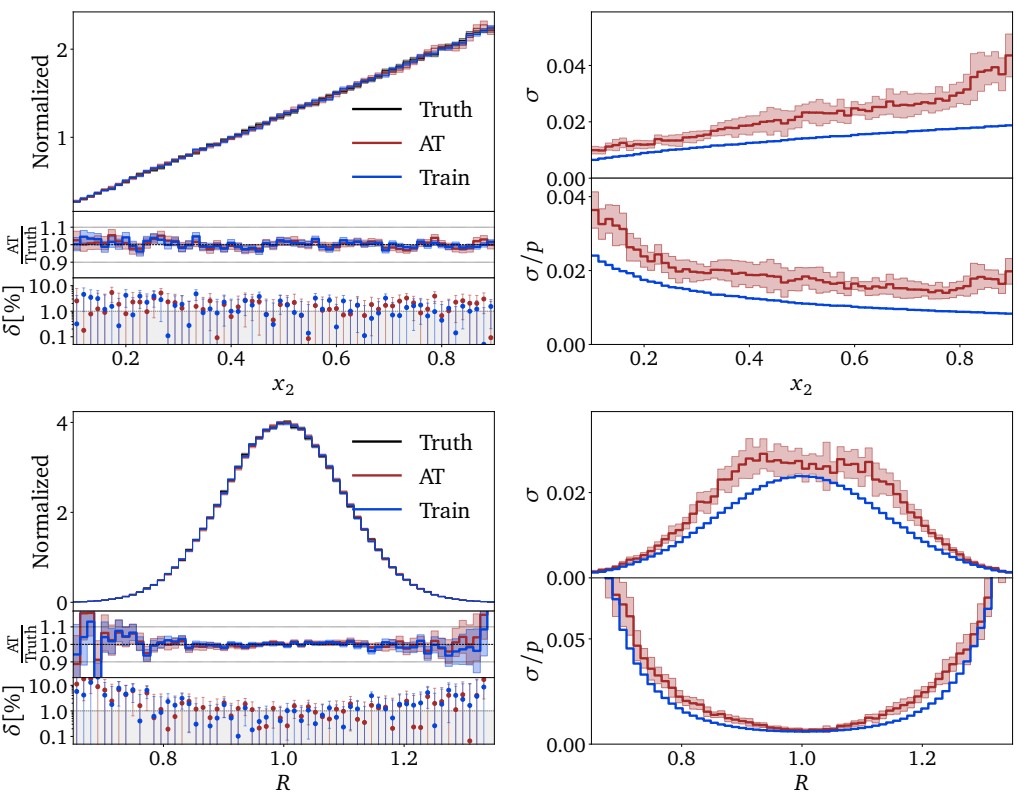

Figure 12: Ramp (upper) and Gaussian ring (lower) distribution from the autoregressive transformer with a Gaussian mixture likelihood. We show the learned density and its Bayesian network uncertainty (left) as well as the absolute and relative uncertainties with a range given by 10 independent trainings, compared to the statistical uncertainty of the training data in blue (right).

autoregressive step, $p_{\text{model}}(x_2|x_1)$, is conditioned on the first component. In the second panel we show $p_{\text{model}}(x_2|x_1 = 0)$ with sharp peaks at $\pm 1$ because the event has to be at the edge of the ring. The Gaussians building the left and right peak are distributed roughly equally. On the other hand, $p_{\text{model}}(x_2|x_1 = 1)$ has a broad plateau in the center, again from the $x_1$-condition.

# 4 LHC events

Most generative network tasks at the LHC are related to learning and sampling phase space densities, for instance event generation at the parton or reconstruction level, the description of detector effects at the reconstruction level, the computation of event-wise likelihoods in the matrix element method, or the inversion and unfolding of reconstructed events. This is why we benchmark our new networks on a sufficiently challenging set of LHC events. Following Ref. [22] we choose the the production of leptonically decaying $Z$-bosons, associated with a variable number of QCD jets,

$$pp \to Z_{\mu\mu} + \{1, 2, 3\} \text{ jets}. \tag{59}$$

The network has to learn the sharp $Z$-peak as well as correlated phase space boundaries and features in the jet-jet correlations. We generate the training dataset of 5.4M events (4.0M + 1.1M + 300k) using SHERPA2.2.10 [87] at 13 TeV, including ISR and parton shower with CKKW merging [88], hadronization, but no pile-up. The jets are defined by FASTJET3.3.4 [89] using the anti-$k_T$ algorithm [90] and applying the basic cuts

$$p_{T,j} > 20 \text{ GeV}, \qquad \text{and} \qquad \Delta R_{jj} > 0.4. \tag{60}$$

The jets and muons are each ordered in transverse momentum. Our phase space dimensionality is three per muon and four per jet, i.e. 10, 14, and 18 dimensions. Momentum conservation is not guaranteed, because some final-state particles might escape for instance the jet algorithm. However, the physically relevant phase space dimensionality is reduced to 9, 13, and 17 by removing the global azimuthal angle.

Our data representation includes a minimal preprocessing. Each particle is represented by

$$\{ p_T, \eta, \phi, m \}. \tag{61}$$

Given Eq.(60), we provide the form $\log(p_T - p_{T,\min})$, leading to an approximately Gaussian shape. All azimuthal angles are given relative to the leading muon, and the transformation into $\text{artanh}(\Delta\phi/\pi)$ again leads to an approximate Gaussian. The jet mass is encoded as $\log m$. Finally, we centralize and normalize each phase space variable as $(q_i - \bar{q}_i)/\sigma(q_i)$ and apply a whitening/PCA transformation separately for each jet multiplicity for the two diffusion models.

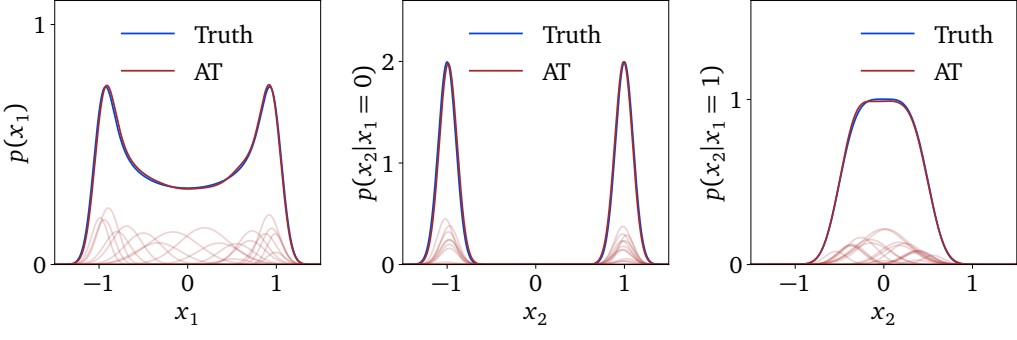

Figure 13: Conditional likelihoods for the Gaussian ring. We show the full Gaussian mixture as well as the 21 individual Gaussians, compared to the truth distribution.

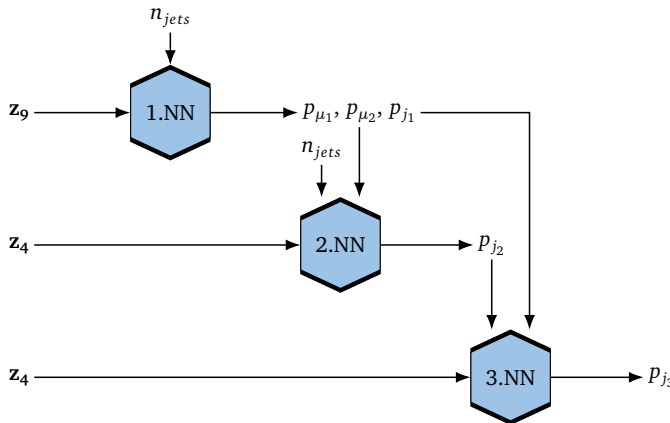

Figure 14: Conditional Sampling Architecture.

## Denoising diffusion probabilistic model

The additional challenge for $Z$+jets event generation is the variable number of jets, which we tackle with a conditional evaluation [22], illustrated in Fig. 14. The training is independent for the three jet multiplicities. We start by giving the information for the $Z + 1$-jet sub-process, 12 phase space dimensions, to a first network. It is supplemented with the one-hot encoded jet count. The second network then receives the 4-momentum of the second jet as an input, and the $Z + 1$-jet information additionally to the jet count as a condition. Analogously, the third network learns the third jet kinematics conditioned on the $Z + 2$-jet information. For democratic jets this conditioning would be perfect, but since we order the jets in $p_T$ it has to and does account for the fact that for higher jet multiplicities the interplay between partonic energy and jet combinatorics leads to differences in the spectra of the leading jets at a given multiplicity.

As discussed in Sec. 2.1 time is a crucial condition for the DDPM network, and we embed it into the full conditioning of the LHC setup as a high-dimensional latent vector linked by a linear layer. We also add a second block to our network architecture, where the conditions are fed to each block individually. The amount of training data is different for the different jet multiplicities and corresponding networks. As shown in Tab. 1, the first network uses the full 3.2M events, the second 850k events with at least two jets, and the third network 190k events with three jets. This hierarchy is motivated by the way the chain of conditional networks add information and also by the increasing cost of producing the corresponding training samples. We could balance the data during training, but for the B-DDPM model this leads to a slight performance drop. We compensate the lack of training data by increasing the number of epochs successively from 1000 to 10000.

Going from toy models to LHC events, we increase the number of blocks to two, which improves the performance. The reason is that we attach the condition to the input at the beginning of each block, so the second block reinforces the condition. Going to even more blocks will slightly improve the performance, but at the expense of the training time.

In Fig. 15 we show a set of kinematic distributions for different jet multiplicities, including the jet-inclusive scalar sum of the up to three $p_{T,j}$. These distributions will be the same for all three network in this paper and can be compared directly to the Bayesian INN results in Fig. 11 of Ref. [22], serving as a precision baseline. Starting with the almost featureless $p_T$-distributions in the left panels, we see that for all three distributions the deviation from the truth, given by high-statistics training data, is similar for the actual training data and for the DDPM-generated events. The network really extracts all available information from the training data combined with its fit-like implicit bias. For sufficient training statistics, the

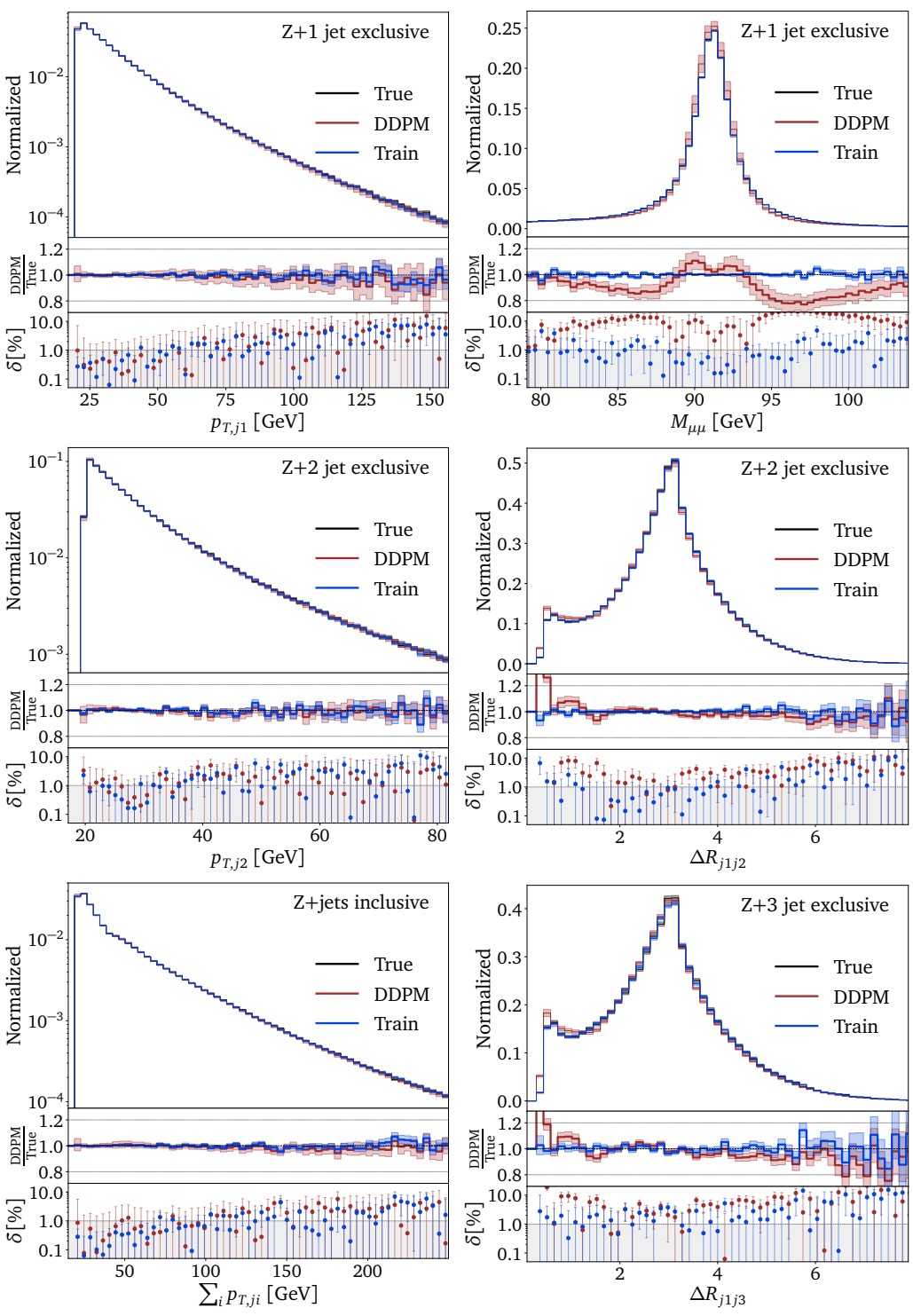

Figure 15: Bayesian DDPM densities and uncertainties for $Z+1$ jet (upper), $Z+2$ jets (center), and $Z+3$ jets (lower) from combined $Z+$ jets generation. The uncertainty on the training data is given by bin-wise Poisson statistics. The network architecture is given in Tab. 1. For a comparison with the INN we refer to Fig. 11 of Ref. [22].

precision on the phase space density as a function of $p_T$ is below the per-cent level, easily on par with the INN baseline. For a given jet multiplicity this precision drops with increasing $p_T$ and correspondingly decreasing training data, an effect that is correctly and conservatively modeled by the uncertainty estimate of the B-DDPM. Combining all $n$-jet samples into one observable is no problem for the network and does not lead to any artifacts.

In the right panels of Fig. 15 we show the most challenging phase space correlations. We start with the $Z$-peak, which governs most of the events, but requires the network to learn a very specific phase space direction very precisely. Here, the agreement between the true density and the DDPM result drops to around 10% without any additional phase space mapping, similar to the best available INN. The deviation is not covered by the Bayesian network uncertainty, because it arises from a systematic failure of the network in the phase space resolution, induced by the network architecture. However, this effect is less dramatic than it initially looks when we notice that the ratio of densities just describes the width of the mass peak being broadened by around 10%. If needed, it can be easily corrected by an event reweighting of the $Z$-kinematics. Alternatively, we can change the phase space parametrization to include intermediate particles, but most likely at the expense of other observables.

Next, we study the leading challenge of ML-event generators, the jet-jet correlations and specifically the collinear enhancement right at the hard jet-separation cut of $\Delta R_{jj} > 0.4$. Three aspects make this correlation hard to learn: (i) this phase space region is a sub-leading feature next to the bulk of the distribution around $\Delta R_{jj} \sim \pi$; (ii) it includes a sharp phase space boundary, which density estimators will naturally wash out; and (iii), the collinear enhancement needs to be described correctly, even though it appears right at the phase space boundary. Finally, for this correlation the conditional setup and the Bayesian extension are definitely not helpful.

What helps for this correlation is the so-called magic transformation introduced in Ref. [22]. It scales the $\Delta R_{jj}$-direction in phase space such that the density in this phase space direction becomes a monotonous function. While from a classic Monte Carlo perspective the benefits of this transformation are counter-intuitive, from a a fit-like perspective the magic transformation can simplify the class of function which the network then adapts to the data, as shown for the toy models in the previous section. This argument is confirmed by the fact that for our diffusion networks this transformation is helpful, just like for the INN, but for the transformer it is not needed. Both, for the 2-jet and the 3-jet sample we see that with the magic transformation the DDPM learns the $\Delta R_{jj}$ features, but at the same 10% level as the INN and hence missing our 1% target. The Bayesian uncertainty estimate increases in this phase space region as well, but it is not as conservative as for instance in the $p_T$-tails.

The challenge of current diffusion networks, also the DDPM, is the evaluation speed. For each additional jet we need to call our network 1000 times, so sampling 3-jet events takes three times as long as sampling 1-jet events. However, none of the networks presented in this study are tuned for generation speed, the only requirement for a limited hyperparameter scan is the precision baseline given by the INN.

### Conditional flow matching

For the CFM diffusion network we follow the same conditional setup as for the DDPM and the INN to account for the variable number of jets. The network is described in Tab. 2, unlike for the DDPM the three networks do not have the same size, but the first network with its 9 phase space dimensions is larger. Also the number of epochs increases from 1000 to 10000 going to the 3-jet network. For the CFM we combine the embedding of the time and the conditioning on the lower jet multiplicities. We find the best results when encoding time, the kinematic condition, and the actual network input separately into same-sized latent vectors with independent linear layers. Then all three are concatenated and given to the network.

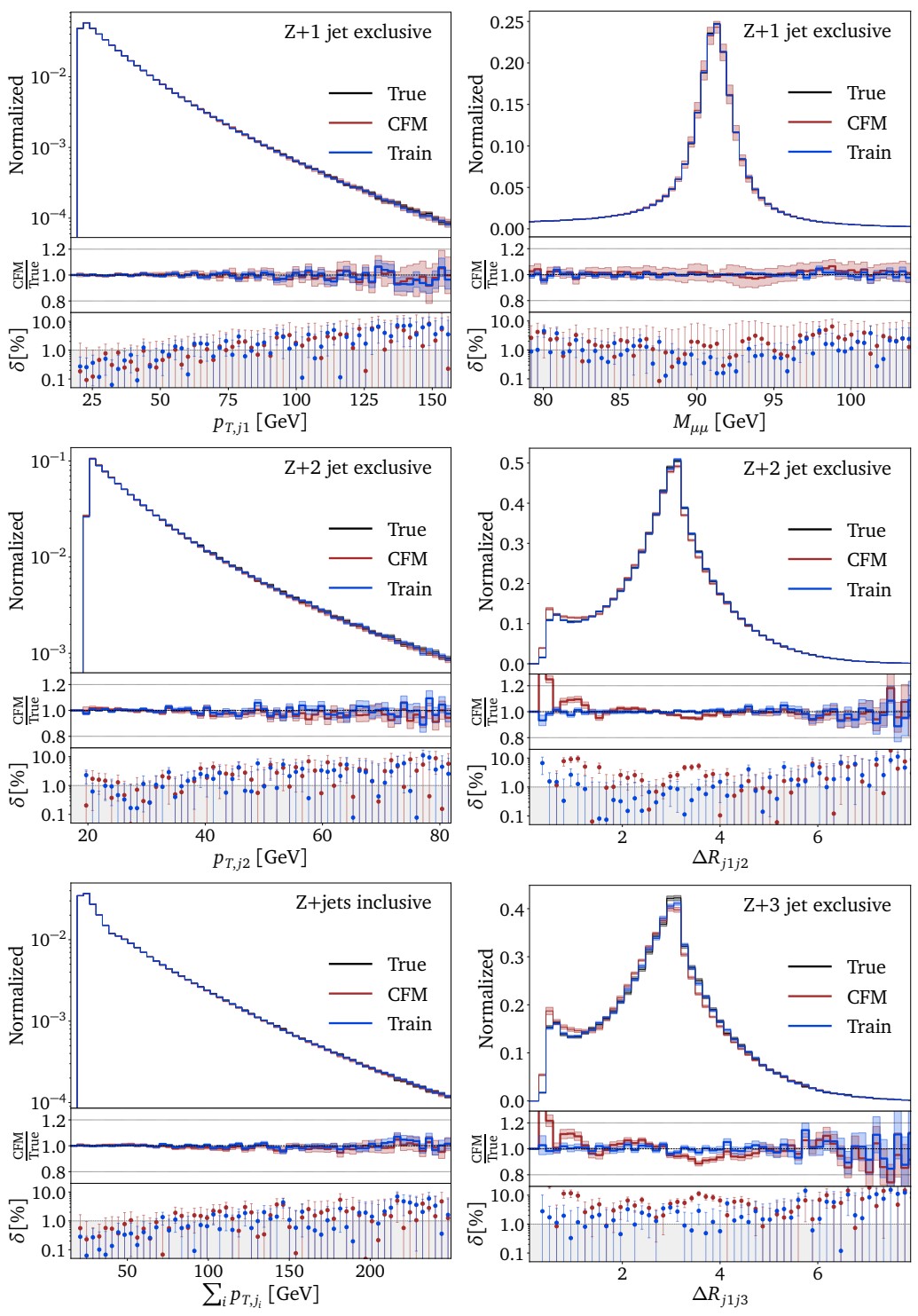

Figure 16: Bayesian CFM densities and uncertainties for $Z + 1$ jet (upper), $Z + 2$ jets (center), and $Z + 3$ jets (lower) from combined $Z+$ jets generation. The uncertainty on the training data is given by bin-wise Poisson statistics. The network architecture is given in Tab. 2. For a comparison with the INN we refer to Fig. 11 of Ref. [22].

The kinematic distributions generated by the CFM are shown in Fig. 16. Again, the transverse momentum spectra are learned with high precision, with decreasing performance in the tails, tracked correctly by the Bayesian network uncertainty. The correlation describing the $Z$-peak is now modeled as well as the bulk of the single-particle distributions, a significant improvement over the INN baseline [22]. For the most challenging $\Delta R_{jj}$ distributions the CFM uses the same magic transformation as the DDPM and achieves comparable precision. This means that while there might possibly be a slight benefit to our CFM implementation with an ODE approach to the discrete time evolution in terms of precision, our level of network optimization does not allow us to attribute this difference to luck vs network architecture. Similarly, in the current implementation the CFM generation is about an order of magnitude faster than the DDPM generation, but this can mostly be attributed to the linear trajectory and the extremely efficient ODE solver.

**Autoregressive transformer**

For the third network, a generative transformer, we already know from Sec. 3 that it learns and encodes the phase space density different from normalizing flows or diffusion networks. A key structural difference for generating LHC events is that the transformer can generate events with different jet multiplicities using the same network. The one-hot-encoded jet multiplicity is provided as an additional condition for the training. The autoregressive structure can work with sequences of different length, provided there is an appropriate way of mapping the sequences onto each other. For the LHC events we enhance the sensitivity to the angular jet-jet correlations through the ordering

$$\left( (\phi, \eta)_{j_{1,2,3}}, (p_T, \eta)_{\mu_1}, (p_T, \phi, \eta)_{\mu_2}, (p_T, m)_{j_{1,2,3}} \right). \tag{62}$$

While the Bayesian transformer does learn the angular correlations also when they appear at the end of the sequence, this ordering provides a significant boost to the network's precision. For the transformer training, we want the features of the 3-jet to be well represented in the set of vectors defined in Eq.(62). To train on equal numbers of events with one, two, and three jets, we sample 1-jet and 2-jet events randomly at the beginning of each epoch. The loss is first evaluated separately for each jet multiplicity, and then averaged for the training update.

In Fig. 17 we show the standard set of kinematic observables for the autoregressive transformer based on a Gaussian mixture model, with the architecture given in Tab. 3. Just like the two diffusion models, and the INN, it learns the different $p_T$-distributions with a precision close to the statistics of the training data. Sampling a variable number of jets with the multiplicity as a condition leads to no additional complication.

Looking at the correlations in the right panels, the $Z$-mass now comes with an increased width and a shift. This is, in part, an effect of the ordering of the input variables, where the lepton information comes after the angular information on the jets. The benefit of this ordering can be seen in the $\Delta R_{jj}$ distributions, which are reproduced at the per-cent precision without any additional effort. This is true for $\Delta R_{j_1 j_2}$ and $\Delta R_{j_1 j_3}$, reflecting the democratic ordering and training dataset. The sharp phase space boundary at $\Delta R_{jj} = 0.4$ can be trivially enforced during event generation.

## 5 Outlook

Generative neural networks are revolutionizing many aspects of our lives, and LHC physics is no exception. Driven by large datasets and precise first-principle simulations, LHC physics

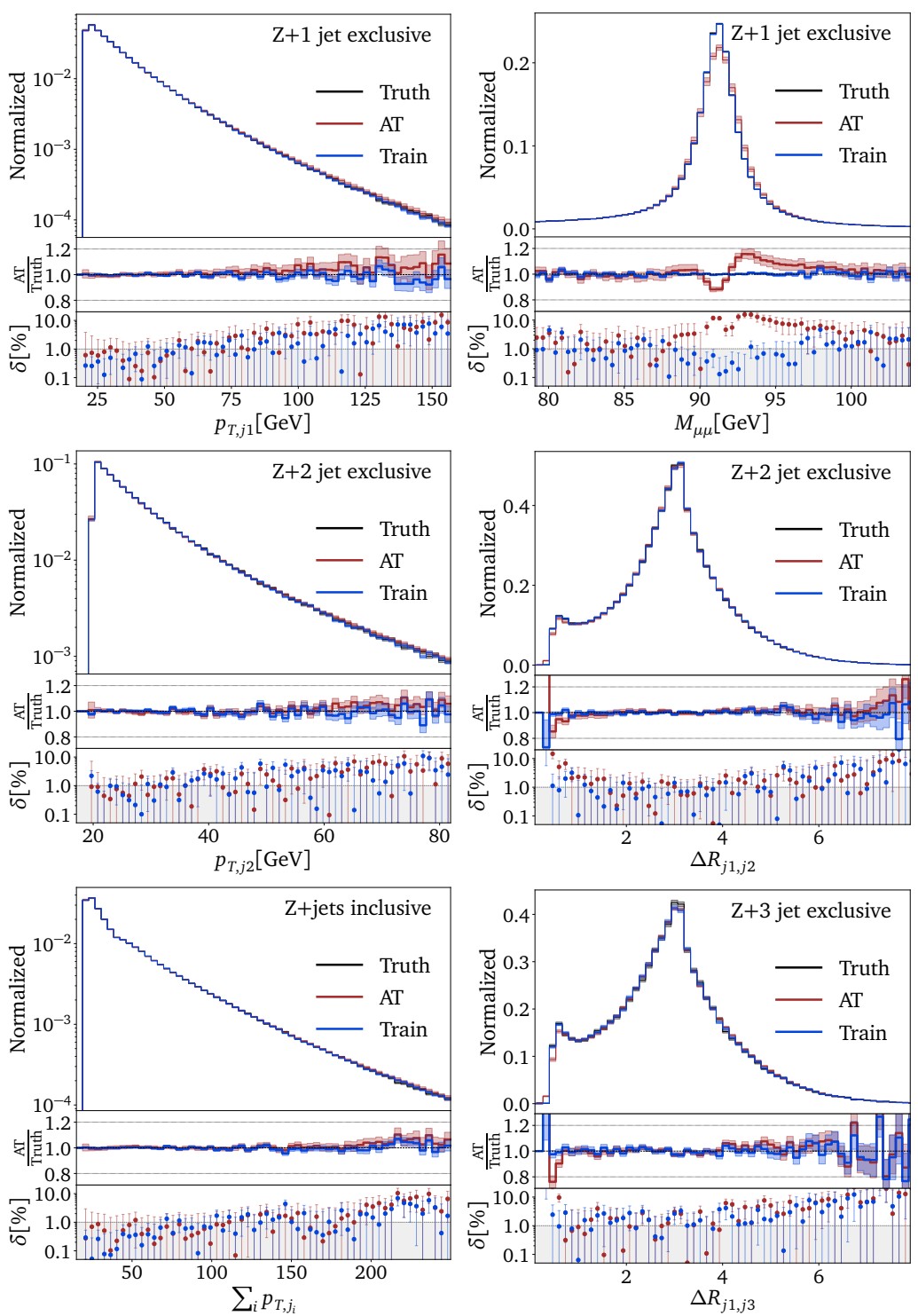

Figure 17: Bayesian autoregressive transformer densities and uncertainties for $Z+1$ jet (upper), $Z+2$ jets (center), and $Z+3$ jets (lower) from combined $Z+$ jets generation. The uncertainty on the training data is given by bin-wise Poisson statistics. The network architecture is given in Tab. 3. For a comparison with the INN we refer to Fig. 11 of Ref. [22].

offers a wealth of opportunities for modern machine learning, in particular generative networks [2]. Here, classic network architectures have largely been surpassed by normalizing flows, especially its INN variant, but cutting-edge new approaches are extremely promising. Diffusion networks should provide an even better balance between expressivity and precision in the density estimation. Autoregressive transformers should improve the scaling of network size and training effort with the phase space dimensionality.

In this paper we have provided the first comprehensive study of strengths and weaknesses of these new architectures for an established LHC task. We have chosen two fundamentally different approaches to diffusion networks, where the DDPM learns the time evolution in terms of discrete steps, while the CFM encodes the continuous time evolution into in a differential equation. The autoregressive JetGPT transformer follows the standard GPT architecture, where for our relatively simple setup we get away without actual pretraining.

For each architecture we have first implemented a Bayesian network version, which allows us to understand the different ways they approach the density estimation. While the diffusion networks first identify classes of functions and then adapt them to the correlations in phase space, much like the INN [69], the transformer learns patterns patch-wise and dimension by dimension.

Next, we have applied all three networks to the generation of $Z$+jets events, with a focus on the conditional setup for variable jet multiplicities and the precision in the density estimation [22]. The most challenging phase space correlations are the narrow $Z$-peak and the angular jet–jet separation combined with a collinear enhancement.

Our two diffusion models are, conceptually, not very far from the INNs. We have found that they face the same difficulties, especially in describing the collinear jet–jet correlation. Just like for the INN, the so-called magic transformation [22] solved this problem. Both diffusion networks provided an excellent balance between expressivity and precision, at least on part with advanced INNs. This included the density estimation as well as the uncertainty map over phase space. The main advantage of the CFM over the DDPM was a significantly faster sampling for our current implementation, at the level of the INN or the transformer. In contrast, the DDPM model is based on a proper likelihood loss, with all its conceptual and practical advantages for instance when bayesianizing it. Both networks required long training, but fewer network parameters than then INN. We emphasize that ML-research on diffusion models it far from done, so all differences between the two models found in this paper should be considered with a grain of salt.

Finally, we have adapted the fundamentally different GPT architecture to LHC events. Its autoregressive setup provided a different balance between learning correlations and scaling with the phase space dimension, and it has never been confronted with the precision requirements of the LHC. The variable numbers of particles in the final state was implemented naturally and without an additional global conditioning. Our transformer is based on a Gaussian mixture model for the phase space coverage, and we have used the freedom of ordering phase space dimensions in the conditioning chain to emphasize the most challenging correlations. This has allowed the transformer to learn the jet–jet correlations better than the INN or the diffusion models, but at the expense of the description of the $Z$-peak. The generation time of the transformer is comparable with the fast INN.

Altogether, we have found that new generative network architectures have the potential to outperform even advanced normalizing flows and INNs. However, diffusion models and autoregressive transformers come with their distinct set of advantages and challenges. Given the result of our study we expect significant progress in generative network applications for the LHC, whenever the LHC requirements in precision, expressivity, and speed can be matched by one of the new architectures.



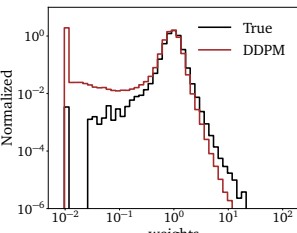 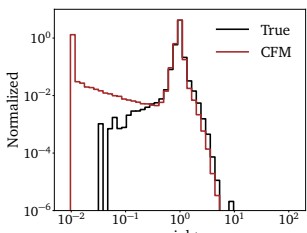 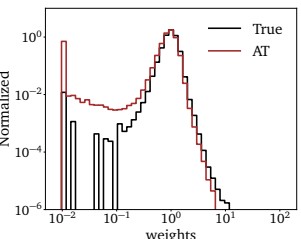

Figure 18: Classifier weight distribution for each of the three networks evaluated on Z+2j events.

# Acknowledgments

We would like to thank Theo Heimel, Michel Luchmann, Luigi Favaro, Ramon Winterhalder, and Claudius Krause for many useful discussions.

**Funding information**  AB, NH, and SP are funded by the BMBF Junior Group *Generative Precision Networks for Particle Physics* (DLR 01IS22079). AB, TP, and JS would like to thank the Baden-Württemberg-Stiftung for financing through the program *Internationale Spitzenforschung*, project *Uncertainties - Teaching AI its Limits* (BWST_IF2020-010). This research is supported by the Deutsche Forschungsgemeinschaft (DFG, German Research Foundation) through Germany's Excellence Strategy EXC 2181/1 – 390900948 (the *Heidelberg STRUCTURES Excellence Cluster*).

# A  Quantitative evaluation of generators

Following Ref. [91] we evaluate the quality of the networks by training binary classifiers to distinguish between generated and true samples. The output $C(x)$ of a well-trained classifier gives access to the likelihood ratio between the true and the model density via

$$w(x) = \frac{p_{\text{true}}(x)}{p_{\text{model}}(x)} = \frac{C(x)}{1 - C(x)}. \tag{A.1}$$

This is done individually for all three models, DDPM, CFM and AT, following the training procedure as discussed in [91]. The corresponding weight distributions are shown in Fig. 18. For all three models the weight distribution shows a similar structure. The overwhelming majority of events is clustered around weights of one, indicating a good agreement between the model and the true distribution. The weak tail to the right shows that no phase space region is systematically underpopulated. Lastly, all three models show a peak in the overflow bin to the left, indicating a clear failure mode. Low values of $w(x)$ correspond to regions where $p_{\text{true}}$ is approaching zero whereas $p_{\text{model}}$ is not. We checked that this excess is due to the already discussed mismodeling around the hard cut-off $\Delta R_{j1j2} = 0.4$.

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
