# Peer review of "Jet Diffusion versus JetGPT -- Modern Networks for the LHC"

_SciPost Physics, doi:SciPost Phys. Core 8, 026 (2025)_

## Round 1 · Author Response

Answer to Report 1

Dear Referee,
Thank you very much for your positive feedback regarding our publication. We agree that there are several interesting studies to improve the understanding of these neural generators. However, we come to the same conclusion as you, that this is beyond the scope of this paper and future work is required to fully cover all of those aspects.
Sincerely, the authors

Answer to Report 2

Dear Referee,
we would like to thank you for the favorable evaluation of our manuscript and for your comments and suggestions for improvement. We addressed all the issues that you raise and believe you will now find the manuscript suitable for a publication in SciPost. For more details please find our remarks/replies threaded into your report.
Sincerely, The authors

>All their plots show pretty good agreement, but only qualitatively. It's hard to tell quantitatively how well the models are working or >where they fail. The paper reads like they just wanted to get these models to work, and they do work ok, but they are not trying to find out >what their relative strenghts and weaknesses. All the comparisons are vague and qualitative.

Thank you for pointing that out. To compare the networks in a more quantitative way, we trained binary classifiers between the simulation and events generated by our networks. We added the corresponding section to the appendix, where we also discuss potential failure modes. Comparing these networks more systematically is challenging, because the relevant metrics depend on the downstream application. The purpose of our study was to show that all three networks are competetive and usable for LHC applications.

>Similarly, they do not clearly state what exactly the problem is with event generation that they are trying to (solve). Is it just speeding >things up? The bottleneck is usually in tails of high multiplicity events, not the peaks. Are they getting those tails right?

We thank you for pointing out this aspect. Indeed the main motivation for using these networks is speed. Generating the training dataset for our experiments took many CPU hours, whereas our models can generate the same number of events in a few minutes. This performance gain will increase further for higher event multiplicity or NLO generators. Multi-jets events tend to be very costly and will be a bottleneck of faithful simulations for the HL-LHC. We do not intend to give a concrete quantitative comparison as this will always be very setup dependent.
Beyond unconditional generation, these models can and have already been used to invert simulation, e.g. in generative unfolding and for the matrix element method. We mention these arguments in the introduction.
Concerning the tails of the distribution, the precision of the networks is naturally limited by a lack of training statistics. This is accurately captured by the uncertainty-aware Bayesian networks. For all models the deviation of the learned density from the underlying truth is covered by the learned uncertainties, even deep in the tails.

---

## Round 1 · List of Changes

• Added further motivation in the 4. paragraph in the introduction
  • Added a section "Quantitative evaluation of generators", where we evaluate the network performances using trained classifiers.

---

## Editorial Decision

published